# Targeting codon 158 p53-mutant cancers via the induction of p53 acetylation

Li Ren Kong [1,2✉], Richard Weijie Ong[3], Tuan Zea Tan [1], Nur Afiqah Binte Mohamed Salleh[1], Matan Thangavelu[4], Jane Vin Chan[4], Lie Yong Judice Koh[4], Giridharan Periyasamy[4], Jieying Amelia Lau[1], Thi Bich Uyen Le[3], Lingzhi Wang[1,5], Miyoung Lee[2], Srinivasaraghavan Kannan [6], Chandra S. Verma[6,7,8], Chwee Ming Lim[9], Wee Joo Chng[1,10,11], David P. Lane [12], Ashok Venkitaraman[2], Huynh The Hung[3], Chit Fang Cheok[13,14,15] & Boon Cher Goh [1,5,10✉]

Gain of function (GOF) DNA binding domain (DBD) mutations of *TP53* upregulate chromatin regulatory genes that promote genome-wide histone methylation and acetylation. Here, we therapeutically exploit the oncogenic GOF mechanisms of p53 codon 158 (Arg[158]) mutation, a DBD mutant found to be prevalent in lung carcinomas. Using high throuput compound screening and combination analyses, we uncover that acetylating mutp53[R158G] could render cancers susceptible to cisplatin-induced DNA stress. Acetylation of mutp53[R158G] alters DNA binding motifs and upregulates TRAIP, a RING domain-containing E3 ubiquitin ligase which dephosphorylates IκB and impedes nuclear translocation of RelA (p65), thus repressing oncogenic nuclear factor kappa-B (NF-κB) signaling and inducing apoptosis. Given that this mechanism of cytotoxic vulnerability appears inapt in p53 wild-type (WT) or other hotspot GOF mutp53 cells, our work provides a therapeutic opportunity specific to Arg[158]-mutp53 tumors utilizing a regimen consisting of DNA-damaging agents and mutp53 acetylators, which is currently being pursued clinically.

[1] Cancer Science Institute of Singapore, National University of Singapore, Singapore 117599, Singapore. [2] Medical Research Council Cancer Unit, University of Cambridge, Cambridge CB2 0XZ, UK. [3] Laboratory of Molecular Endocrinology, National Cancer Centre Singapore, Singapore, Singapore. [4] Genome Institute of Singapore, Agency for Science, Technology & Research (A*STAR), Singapore 138672, Singapore. [5] Department of Pharmacology, Yong Loo Lin School of Medicine, National University of Singapore, Singapore 117597, Singapore. [6] Bioinformatics Institute, Agency for Science, Technology, and Research (A*STAR), Singapore 138671, Singapore. [7] Department of Biological Sciences, National University of Singapore, Singapore 117558, Singapore. [8] School of Biological Sciences, Nanyang Technological University, Singapore 639798, Singapore. [9] Division of Surgical Oncology (Head and Neck Surgery), National University Cancer Institute, Singapore (NCIS), Singapore 119074, Singapore. [10] Department of Haematology-Oncology, National University Cancer Institute, Singapore 119074, Singapore. [11] Department of Medicine, Yong Loo Lin School of Medicine, National University of Singapore, Singapore 119228, Singapore. [12] p53 Laboratory (p53Lab), Agency for Science, Technology, and Research (A*STAR), Singapore 138648, Singapore. [13] Institute of Molecular and Cell Biology (IMCB), Agency for Science, Technology, and Research (A*STAR), Singapore 138673, Singapore. [14] Department of Pathology, Yong Loo Lin School of Medicine, National University of Singapore, Singapore 119074, Singapore. [15] Department of Biochemistry, Yong Loo Lin School of Medicine, National University of Singapore, Singapore 117596, Singapore. ✉email: csiklr@nus.edu.sg; phcgbc@nus.edu.sg

TP53 missense mutations are among the most common genetic lesions in tumors[1], which often coincide with the earlier onset of oncogenesis than patients with p53 loss[2]. A single nucleotide substitution at the DNA-binding domain (DBD) renders the protein defective in DNA-binding, loss of tumor suppressive properties and concomitantly prevents the negative feedback regulation through MDM2[3,4], leading to massive accumulation of full length mutant p53 (mutp53). Growing evidence from recent studies suggest that cells with prevalent mutp53 acquire additional oncogenic gain-of-function (GOF) based on their unique structural modifications[5–8]. Depletion of mutp53 or inhibition of its co-activator have demonstrated strong cytotoxicity in tumor cells[6,9,10].

Proposed oncogenic mechanisms of hotspot p53 mutations include prolonged tumor necrosis factor alpha (TNF-α) signaling through the activation of NFκB (nuclear factor kappa-light-chain-enhancer of activated B cells)[11,12], causing chronic tumor-associated inflammation, as well as altered structural interaction between mutated p53 and DNA that induces transcriptional perturbations to promote tumor-associated gene expression[13–15]. Data derived from The Cancer Genome Atlas (TCGA) reveal a specific point mutation on arginine codon 158 (Arg[R158]) to be a recurrent mutation in lung carcinomas (16 out of 742 specimens)[16–19]. In contrast to the other well-established hotspot mutp53[7,8,20–23], the functional aspects of this mutation have not been well-characterized.

In this study, we uncover a mechanism of activating mutp53-dependent apoptotic function in cancer cells through p53[R158G] acetylation, and demonstrate that TRAIP regulation of NFκB is the main molecular driver underpinning this observed sensitivity. We further show in a high-throughput screen that acetylation of p53[R158G] can be achieved with several pharmacologic agents, providing a cogent basis for further clinical development.

## Results

**GOF p53[R158G] confers differential drug sensitivity.** Among the TP53 mutations found in ~50% of non-small cell lung cancer[24], p53[R158G/H/L] is one of the most common mutation hotspots according to multiple public databases (TCGA, COSMICS, IARC p53 Database), despite being reported in different frequencies[25]. Further TCGA analysis on sequencing of 742 lung cancer patients showed a frequency of 4.5% (n = 8) and 1.4% (n = 8) in LUSC (n = 178) and adenocarcinoma (LUAD, n = 564) subtypes, respectively (Supplementary Fig. 1A, Supplementary Table 1). In addition, the frequency of codon 158 was found to be comparable to other hotspot mutp53 in both lung cancer subtypes (Supplementary Table 2). H2170 LUSC cells that are homozygous for p53[R158G] (Supplementary Fig. 1B) demonstrated impaired MDM2 and CDKN1A transactivation when treated with Nutlin-3a, a MDM2 antagonist, as compared to MRC5 (p53[wt]) cells, indicating loss of p53 function (Supplementary Fig. 1I). To gain better insights into the p53[R158G] function, we generated isogenic cell-lines expressing either wild-type (p53[wt]) or mutant (p53[R158G]) p53 from homozygous deleted LUSC Calu-1 cells (p53[−/−]). As forced expression of WT p53 could induce cytotoxicity, we verified the presence of full length TP53 in each isolated stable clones (Supplementary Fig. 1C–H). As expected, expression of wild-type p53 (wtp53) increased transcription of MDM2, CDKN1A, PUMA, and PMAIP1 transcripts compared to p53[−/−] cells; in p53[R158G] cells, elevated CDKN1A showed partial preservation of p53 function, but reduced PMAIP1 transcription indicated gain of alternative function (Supplementary Fig. 1J–M). Functionally, mutp53[R158G] overexpression significantly increased cellular motility (Fig. 1a, b) as well as anchorage-independent colony formation (Fig. 1e, f); whereas invasiveness of H2170 cells

could be reduced with TP53 knockdown (Fig. 1c, d). In contrast, overexpression of wtp53 exerted strong tumor suppressive effects in Calu-1 cells by reducing invasiveness (Fig. 1a, b) with no apparent colony growth. Importantly, xenograft tumors derived from p53[R158G] cells demonstrated more aggressive growth relative to those from p53[−/−] and p53[wt] cells (Fig. 1g, h), consistent with the oncogenic GOF described in other hotspot variants[10,22,26].

Understanding the mechanisms underlying mutp53 GOF has allowed for anti-tumoral strategies aimed at inhibiting the oncogenic attributes of mutp53[6,9,10,27–29]. Accordingly, we screened isogenic Calu-1 cell-lines through high-content compound screening (Supplementary Table 3), and quantifying the relative cell viability after single drug treatment at 0.1 or 1 μM, respectively. We first selected for compounds that showed efficacy (> 50% growth inhibition) in at least one cell type (Supplementary Table 3), and identified inhibitors specific to p53[−/−], p53[wt], and p53[R158G] cell-lines (Fig. 2a, Supplementary Fig. 2A, B). Loss of p53 was found to confer sensitivity to paclitaxel as suggested previously (Supplementary Fig. 2A)[30]. Interestingly, 17 compounds demonstrated selective sensitivity in p53[R158G] cells compared with wtp53 and null cells (Fig. 2a, Supplementary Fig. 2A, B). Among these, potent inhibitory effects (10-fold differences in ED_{50} values) against p53[R158G] cells were confirmed in bleomycin, AZD7762, cladribine, topotecan, nocodazole, volasertib, 17-AAG, belinostat (PXD101), bosutinib, and JQ1 using dose-response evaluation (Supplementary Fig. 2C). Consistent with previous reports[9,10,31], inhibitors of HSP90, histone deacetylase (HDAC) and polo-like kinase (PLK) demonstrated selective activity against mutp53 cells.

HDAC inhibitors have demonstrated synergistic activities through epigenetic mechanisms with a variety of clinically approved agents in several cancers. Accordingly, the clinically approved belinostat[32] was selected to delineate the underlying mechanism conferring sensitivity to mutp53. We performed a combination screening on belinostat with cisplatin, a DNA-damaging cytotoxic agent that forms the backbone of chemotherapy against metastatic SCC. In this context, the cell viability (%) and combinatorial score of a belinostat/cisplatin combination was quantified in our study models (Fig. 2b, c). The combination was antagonistic on p53[wt] cells (Bliss value > 0), possibly due to concurrent activation of cell cycle checkpoints[33,34]. In contrast, profound synergistic response was observed in p53[R158G] cells (Bliss value < 0) when combining belinostat (0.1 μM) with cisplatin (0.1–10 μM), indicating that the observed cell response was greater than the predicted values (Fig. 2c). Concomitantly, while belinostat lowered cisplatin IC_{50} in mutp53[R158G] Calu-1 cells, it had little effect on cell-lines with wild-type or null p53 status (Fig. 2d). The enhanced cytotoxicity was validated in H2170 cells, with belinostat augmenting cisplatin-induced caspase 3 and PARP cleavage (Fig. 3a). The fold increase of cleaved PARP and caspase 3 were compared across cisplatin treatment and cisplatin/belinostat combination to indicate presence of synergy (Fig. 3b). Interestingly, similar observation was observed in other cancer types harboring Arg[158] p53 mutations, including H441 (lung adenocarcinoma, p53[R158L]), H661 (lung large cell carcinoma, p53[R158L]), and H747 (colorectal carcinoma, p53[R158L]) (Fig. 3a, b). On the contrary, cancer cells expressing other hotspot GOF p53 mutations, such as LUSC cells H596 (p53[G245C]) and ChaGo-k-1 (p53[C275F]); lung adenocarcinoma H1417 (p53[R175L]) and H1975 (p53[R273H]); breast carcinoma SK-BR-3 (p53[R175H]), HCC70 (p53[R248Q]), BT-549 (p53[R249S]) and MDA-MB-468 (p53[R273H]); pancreatic carcinoma MIA-Paca-2 (p53[R248W]) and PANC-1 (p53[R273H]); did not show increased in apoptotic markers when belinostat was combined with cisplatin (Fig. 3b, Supplementary Fig. 3A, B).

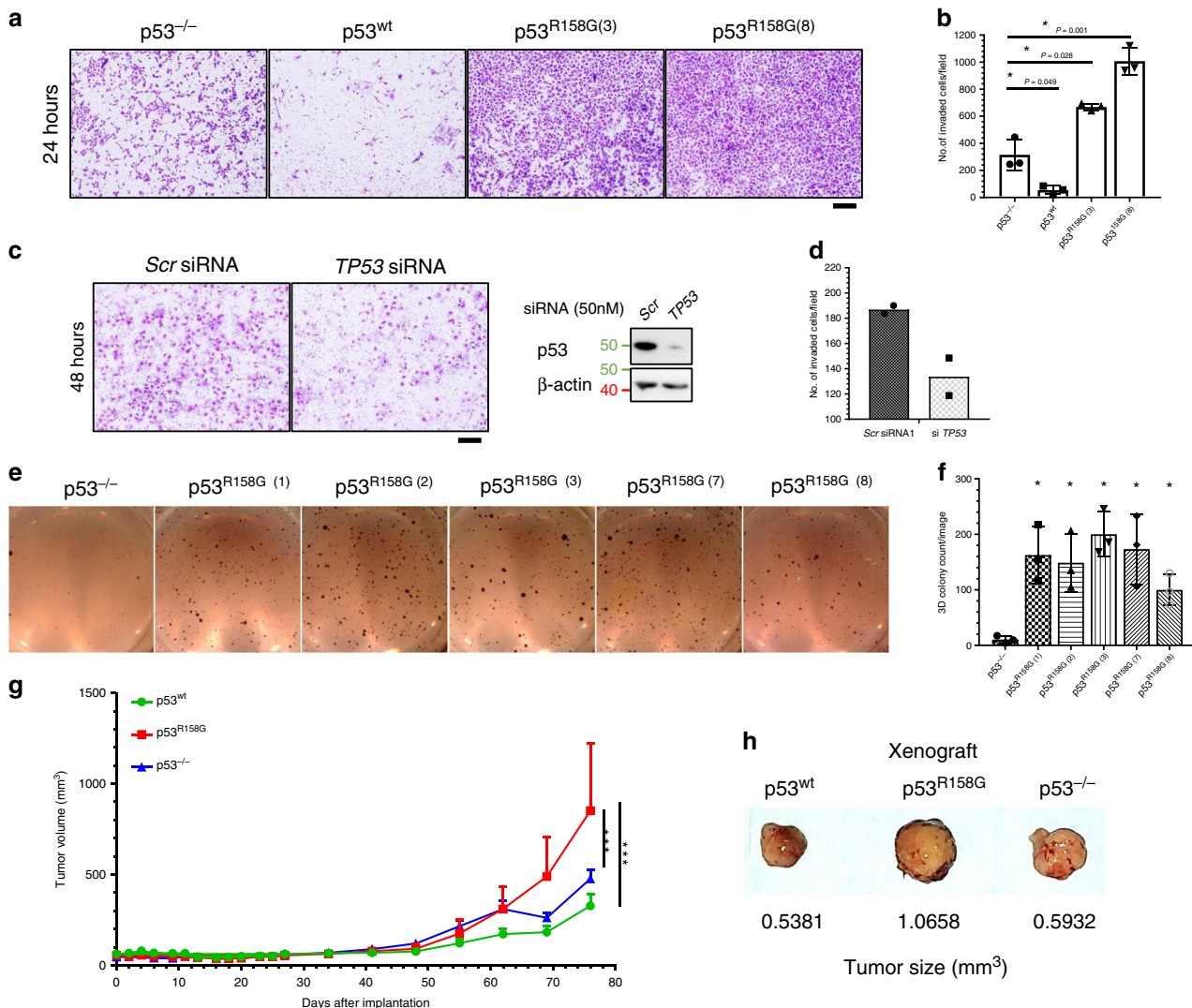

**Fig. 1 Mutation at Arg[158] is a GOF p53 isoform. a–d** Cell invasion assays were performed on isogenic Calu-1 cells (p53[−/−], p53[wt] and mutp53[R158G]) and H2170 cells. Cells seeded in Matrigel invasion chambers were fixed and stained at the indicated time point. Representative images were shown for Calu-1 clones ($n = 3$ independent experiments) (**a**) and H2170 cells with *Scr* of *TP53* siRNA knockdown ($n = 2$ independent experiments) (**c**). Immunoblots verifying p53 knockdown are shown on the right. The number of cells in four random microscopic fields ×4 were quantified for Calu-1 (**b**) and H2170 (**d**) cells. Data are expressed as mean ± SD. 50 nM of siRNA used per transfection. Two tailed Student's *t*-test; *$P < 0.05$. Scale bar, 200 μm. **e, f** Anchorage-independent colony forming assays were performed on isogenic Calu-1 cells (p53[−/−] and five independent mutp53[R158G] clones). Representative images of colony formation for each cell type, stained with MTT at assay endpoint (**e**). The number of colonies were quantified (**f**) and expressed as mean of triplicates ± SD ($n = 3$ independent experiments). Two tailed Student's *t*-test; *$P < 0.05$. **g, h** In vivo growth rates of p53[−/−], p53[wt], p53[R158G] cells were measured upon day of implantation and monitored over time (**g**). Images shown are representative tumor sizes at assay endpoint (**h**). Tumor sizes are presented as mean ± SEM ($n = 5$ animals). Two-way ANOVA with Bonferroni correction; *$P < 0.05$; **$P < 0.01$; ***$P < 0.001$.

Similar lack of synergism was observed in other lung cancer cells with p53 deletion (Calu-1), point-nonsense mutation (H520, SK-MES-1) and wild-type p53 (A549) (Fig. 3b, Supplementary Fig. 3A). These findings indicate specific mechanistic underpinnings of the observed synergy in Arg[158] mutp53 cells.

**Acetylation of mutp53[R158G] induced pro-apoptotic phenotype.** As HDAC inhibitors can acetylate histones and other cellular proteins, possible mechanisms that modulate mutp53 function include Lysine acetylation of histones affecting DNA configuration favoring mutp53 access to DNA-binding elements, or direct modification of mutp53 itself leading to alternative DNA binding. To evaluate this, we first demonstrated that a low, sub-cytotoxic concentration of belinostat (0.1 μM) was sufficient to acetylate both H3 and H4 (Fig. 3c) in H2170 cells, and was synergistic with

cisplatin in apoptotic assays (Fig. 3d, f–h; Supplementary Fig. 3C, D). The transcriptional regulation of p53 targets (*CDKN1A, PUMA, PMAIP1*, BAX) and Bcl-2 family members (*BCL2, BAD, BAK1*) strongly implicated p53 transcriptional regulation in this observed synergy with DNA-damaging cisplatin (Fig. 3j, k). The dependence on mutp53[R158G], functioning through active transcription regulation, was further substantiated by the positive p53-reporter-luciferase activity of combination drug-treatment (Supplementary Fig. 4A). In addition, DNA-binding affinity of mutp53 was validated with gel retardation assay, with wtp53 and mutp53 showing comparable binding affinity to a 31-bp p53 response element sequence from *CDKN1A* (Supplementary Fig. 4B, C). Consistently, depletion of p53 from H2170 parental cells with different short hairpin (shRNA) constructs or small interfering RNA (siRNA) reduced PARP and caspase 3 cleavage

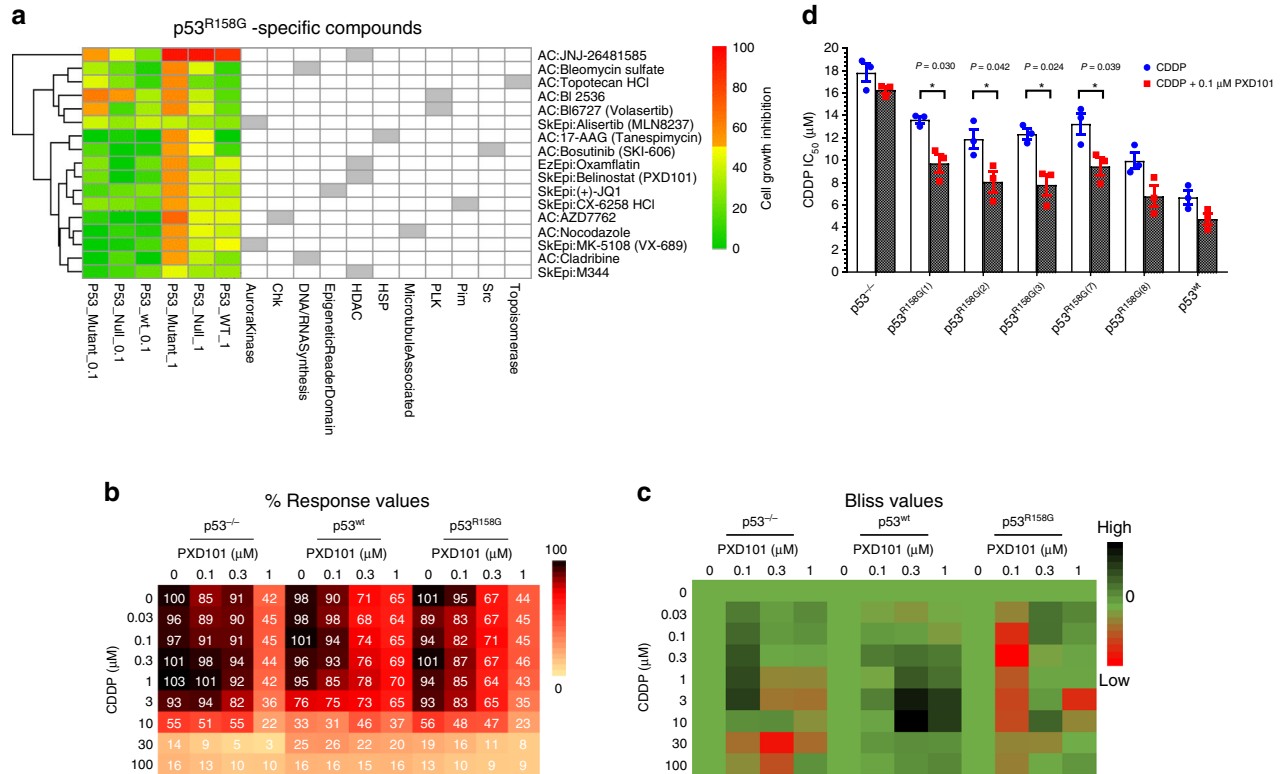

**Fig. 2 High-content screening reveals compounds selective for p53$^{R158G}$ cells. a** High-content screening of anti-cancer compounds and epigenetic modulators on p53 clones. The efficacy of the tested compounds was quantified relative to the mean viability of vehicle-treated cells (384-well format, in triplicates) 72 h post-treatment ($n = 1$). Heatmap shows compounds filtered for specificity to p53$^{R158G}$ cells (growth inhibition > 50% in p53$^{R158G}$; <50% in p53$^{-/-}$, p53$^{wt}$). **b, c** Response profile for belinostat (PXD101) and cisplatin (CDDP). Percent response values represent normalized growth relative to vehicle control (**b**). Data are presented as mean cell viability ($n = 3$ independent experiments). Viability response of belinostat/cisplatin combination from **b** was evaluated using the Bliss combination model (**c**). Bliss Scores are shown, with lower scores (synergy) highlighted in increasingly sharper shades of red. High Bliss values (CI > 0) indicate antagonism; low values (CI < 0) indicate synergy ($n = 3$ independent experiments). **d** Cisplatin IC$_{50}$, with or without belinostat (0.1 μM), was tabulated for Calu-1 isogenic cells (p53$^{-/-}$, p53$^{wt}$, p53$^{R158G}$). Cell viability was measured with MTS assay 72 h post-treatment. Data are represented as mean IC$_{50}$ ± SD ($n = 3$ independent experiments). Two tailed Student's $t$-test; *$P < 0.05$.

(Supplementary Fig. 4D). Apoptosis induced by the combination treatment was significantly abrogated in the shp53 cells (Supplementary Fig. 4E). Logically, destabilization and depletion of GOF mutp53 explains potent anti-tumor cytotoxicity[9,10], but paradoxically, we observed no reduction of mutp53$^{R158G}$ in H2170 cells despite robust apoptosis (Fig. 3e). Rather, analyses of the post-translational modifications (PTM) of p53 showed that while cisplatin dose-dependently increased p53 phosphorylation and acetylation, combination with belinostat potentiated only its acetylation (Fig. 3e).

To establish the role of p53$^{R158G}$ in promoting apoptosis, we conducted synonymous experiments using isogenic Calu-1 cell-lines. Cisplatin alone and in combination with belinostat triggered comparable S-phase arrest regardless of the p53 status, suggesting that the synergy is independent of its cytostatic effect (Figs. 3i, 4c). Cisplatin alone induced pronounced apoptotic response in both p53$^{wt}$ and p53$^{R158G}$ cells, however, enhanced cytotoxicity and p53 acetylation by belinostat were restricted to p53$^{R158G}$ cells (Fig. 4a, b, d), which is concordant with the IC$_{50}$ values (Fig. 2d). Transcriptional activation of *MDM2* was only observed in wtp53 but not mutp53 or null cells (Fig. 4e); whereas, *CDKN1A*, *PUMA*, and *BAX* were induced post-treatment in mutp53 but not null cells (Fig. 4f–i), implicating p53$^{R158G}$ in the synergistic pro-apoptotic effect.

To investigate the role of p53 acetylation in mediating the pro-apoptotic function of mutp53, we first constructed p53$^{R158G}$ mutants by substituting the Lysine residues at the DBD

(p53$^{R158G(DBD, K-A)}$), C-terminal domain (p53$^{R158G(CT, K-A)}$) or the whole protein (p53$^{R158G(K20A)}$). Expectedly, these modifications were unable to overcome cell death induced by single agent cisplatin, which could occur via the p53-independent extrinsic apoptotic pathway[35]. Nonetheless, belinostat-induced synergy was abrogated in p53$^{R158G(DBD, K-A)}$ and p53$^{R158G(K20A)}$ cells (Fig. 4a, b, d), both cell types with impaired Lysine acetylation at the p53 DBD. In contrast, the synergistic effect was maintained in p53$^{R158G(CT,K-A)}$ clones (Fig. 4b), suggesting that C-terminal acetylation is not involved in mutp53-effected cytotoxicity.

To compare the impact of acetylation and phosphorylation on p53$^{R158G}$-dependent apoptosis, we conducted high-content immunofluorescence imaging to detail the post-translational events in individual nuclei (Fig. 4j), and two key observations were made. Firstly, single treatment of belinostat and cisplatin increased the acetylated-p53 population (Q4), with the combination treatment further augmented cells in Q4 (Supplementary Fig. 5A, C). Secondly, cisplatin but not belinostat treatment strongly increased the phosphorylated-p53-labelled cell population (Q3), and combination treatment had no effect on cells in Q3 (Supplementary Fig. 5B, D). This data demonstrated the positive correlation of acetylated p53$^{R158G}$ with combination treatment, which significantly triggered cellular apoptosis. To further validate the importance of p53$^{R158G}$ acetylation, isogenic p53 cell-lines were treated with tenovin-6, a potent sirtuin inhibitor and known acetylator of p53[36]. While p53-expressing cells were more sensitive to tenovin-6 (Fig. 4k), acetylation of p53$^{R158G}$ increased cleavage

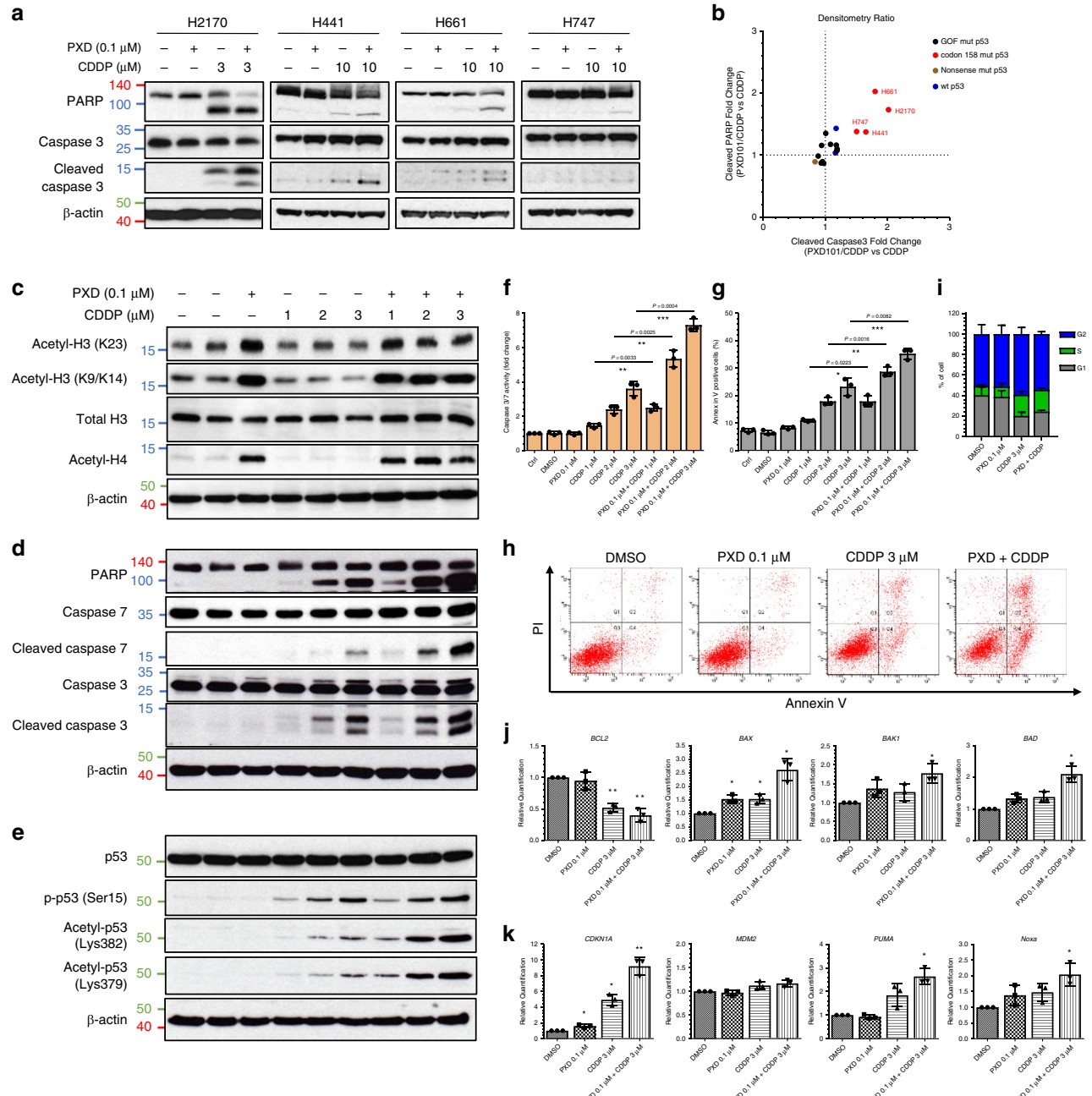

**Fig. 3 Synergistic cytotoxicity of cisplatin and belinostat in carcinoma cells with Arg[158] mutp53 status. a** Western blot measuring the changes in PARP and caspase-3 in Arg[158] mutp53 cells (H2170, H441, H661, and H747) after 48 h treatment with belinostat (PXD101; 0.1 μM) and cisplatin (CDDP; 3 or 10 μM). β-actin shown as loading control (*n* = 3 independent experiments). **b** Densitometry of cleaved PARP (*x*-axis) and caspase 3 (*y*-axis), respectively, for blots in Fig. 3a and Supplementary Fig. 3A, B. Data was quantified and normalized to β-actin, tabulated as ratio of belinostat/cisplatin combination to cisplatin alone (**b**). Data are represented as mean of three independent experiments. **c-e** H2170 cells were treated with increasing concentrations of cisplatin (1, 2, 3 μM) in combination with belinostat (0.1 μM) for 48 h. Western blot indicating the changes in endogenous histones (**c**), apoptotic markers (**d**), and post-translational modifications of p53 (**e**). β-actin shown as loading control (*n* = 3 independent experiments). **f-i** Effects of the indicated treatments on caspase 3/7 activation (**f**), apoptosis (Annexin V staining) (**g, h**) and cell cycle (PI staining) (**i**) were quantified after 48 h. Data are represented as mean ± SD (*n* = 3 independent experiments). Two tailed Student's *t*-test; *P < 0.05, **P < 0.01, ***P < 0.001. **j-k** RT-qPCR analyses of the changes in mRNA levels of apoptotic markers (*BCL2, BAX, BAK1, BAD*) (**j**) and p53 downstream targets (*CDKN1A, MDM2, PUMA, PMAIP1*) (**k**) in H2170 cells 48 h post-treatment. Data are represented as mean ± SD (*n* = 3 independent experiments). Two tailed Student's *t*-test; *P < 0.05, **P < 0.01.

of PARP and caspase 3 (Fig. 4l). Collectively, acetylation of the DBD is crucial to effecting apoptosis by mutp53[R158G].

**Mutp53[R158G] binds to distinct DNA sequence motifs.** We postulated that mutp53[R158G] binds to and activates a distinct

spectrum of chromatin regulatory genes from wtp53, which may be significantly altered by drug-induced acetylation. Firstly, state-of-the-art molecular modelling and simulation studies were performed to compare the structural dynamics of wild-type p53, mutp53[R158G] and acetylated-mutp53[R158G] using monomers of the respective DBD. The loss of the R158 sidechain creates a large

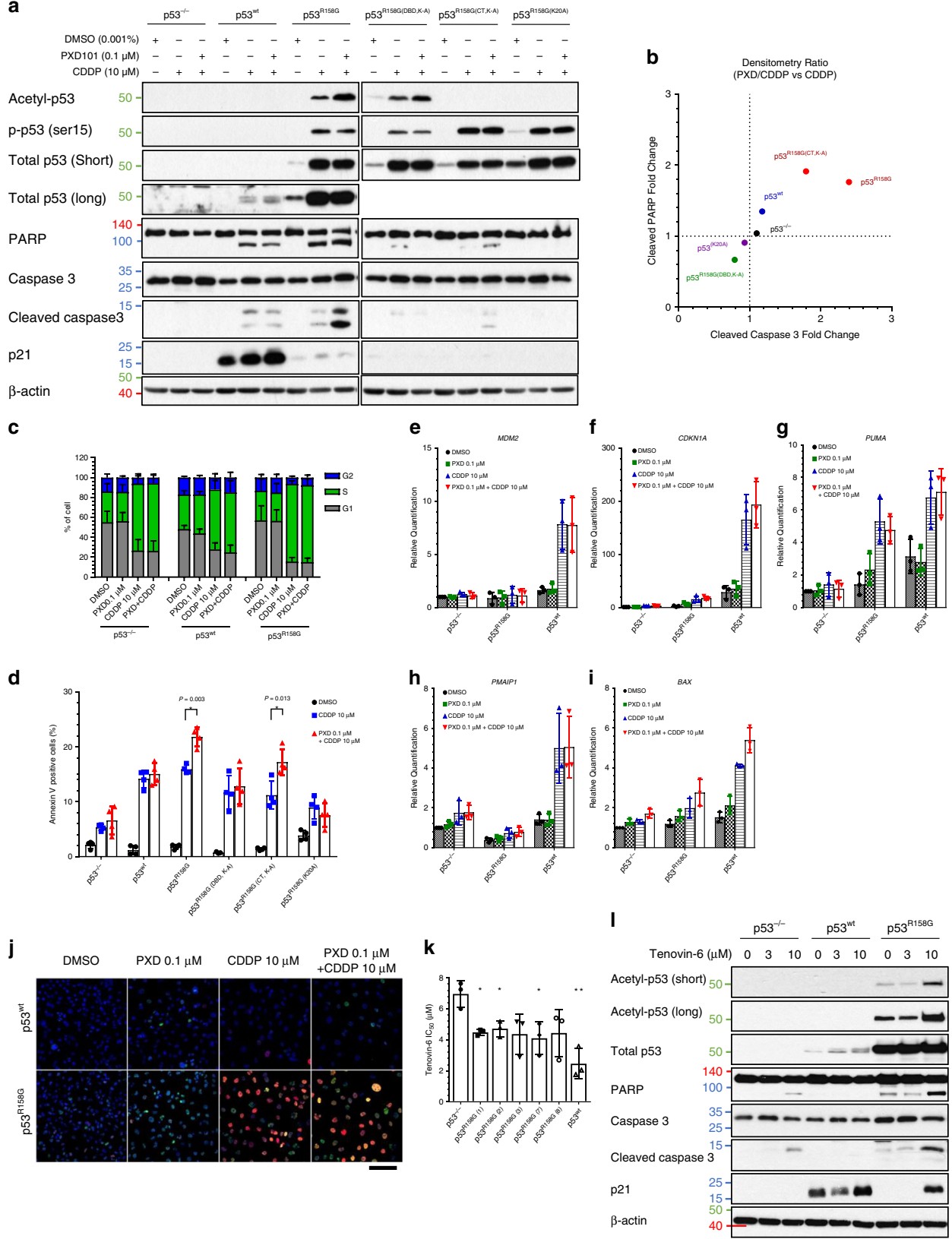

cavity that changes the structure of the DBD – most notable is the loss of the stability of helix H1, which is responsible for dimer formation of activated p53 (Supplementary Fig. 6A, C)—that could likely disrupt key interactions between the DBD (K120, R280, R248) and DNA. Interestingly, acetylation of the Lysine residues within the DBD reduces the sidechain positive charges, resulting in structural and chemical alterations, which suggest restoration of the dimerization capacity of the core domain of mutp53$^{R158G}$ (Supplementary Fig. 6B, D). While the key interactions of K120 with the major groove of the DNA remain

**Fig. 4 Acetylation of mutp53 at the DBD is required for the induction of p53-dependent apoptosis. a, b** Western blot measuring the changes in the indicated protein after 48 h treatment with belinostat (PXD101; 0.1 μM) and cisplatin (CDDP; 10 μM) in Calu-1 isogenic cells generated through site-directed mutagenesis (p53$^{-/-}$, p53$^{wt}$, p53$^{R158G}$, p53$^{R158G(DBD, K-A)}$, p53$^{R158G(CT, K-A)}$, p53$^{R158G(K20A)}$) ($n = 3$ independent experiments) (**a**). β-actin shown as loading control. Densitometry of cleaved PARP (x-axis) and caspase 3 (y-axis) was quantified and normalized to β-actin, tabulated as ratio of belinostat/cisplatin combination to cisplatin alone (**b**). Data are represented as mean of three independent experiments. **c, d** Cell cycle analysis (PI staining) (**c**) and apoptotic profile (Annexin V staining) (**d**) were evaluated 48 hours post-treatment. Data are represented as mean IC$_{50}$ ± SD ($n = 4$ independent experiments). Two tailed Student's t-test; *$P < 0.05$. **e–i** RT-qPCR analyses of the changes in mRNA levels of *MDM2* (**e**), *CDKN1A* (**f**), *PUMA* (**g**), *PMAIP1* (*Noxa*) (**h**), *BAX* (**I**) in p53$^{-/-}$, p53$^{wt}$, p53$^{R158G}$ cells 48 h post-treatment. Data are represented as mean ± SD ($n = 3$ independent experiments). **j** Immunofluorescence staining was performed to determine the colocalization of acetyl-p53 (Lys382) (Alexa Fluor-488) and p-p53 (Ser15) (Alexa Fluor-568) in p53$^{wt}$ and p53$^{R158G}$ cells. Eight independent fields were taken for each condition with a minimum of 50 nuclei per field. Representative confocal images shown at ×40 magnification ($n = 3$ independent experiments). Merged images displayed with blue indicates DAPI, green indicates acetyl-p53, red indicates p-p53, whereas yellow indicates colocalization of p- and acetyl-p53. Scale bar, 50 μm. **k** Tenovin-6 IC$_{50}$ was quantified in Calu-1 isogenic clones 72 h post-treatment. Data are represented as mean IC$_{50}$ ± SD ($n = 3$ independent experiments). Two tailed Student's t-test; *$P < 0.05$; **$P < 0.01$. **l** Western blot indicating changes in the indicated proteins in p53$^{-/-}$, p53$^{wt}$, p53$^{R158G}$ cells 48 h after vehicle or tenovin-6 treatment (3 or 10 μM). β-actin shown as loading control ($n = 3$ independent experiments).

impaired in the acetylated R158G mutant, it is compensated for by the interactions between acetylated K201 and the minor groove (Supplementary Fig. 6D). The most intriguing alteration in the acetylated R158G mutant is an increase in the length of the dimerization helix H1 from R181 to D186—about 1.5 extra turns of a helix—that will promote the formation of a dimer that is more stable than the WT and is also associated with the introduction of two salt bridges across the dimer interface (E180 of each monomer forms a salt bridge with R181 of the second monomer; Supplementary Fig. 6D). We speculate that this may increase DNA binding and result in the enhanced transcription activity seen in the acetylated mutp53$^{R158G}$. The simulations also highlighted the formation of key stabilizing interactions between the polar groups introduced by the acetylated K101 and K164 that stabilize the overall conformation of the mutp53. This supports our earlier observations that conversions of these two Lysines to Alanines in the DBD (p53$^{R158G(DBD,K-A)}$) resulted in attenuation of the pro-apoptotic activity seen in acetylated R158G (Fig. 4a, b, d).

Next, we compared the DNA-binding sequences and transcriptional targets of vehicle- and drug-treated p53$^{wt}$ and p53$^{R158G}$ cells through p53 chromatin immunoprecipitation (ChIP) followed by sequencing (ChIP-Seq). In untreated cells, both wtp53 and mutp53 bound significantly to the proximal regions (within 1 kb) of the transcription start sites (TSS) (Fig. 5a, Supplementary Fig. 7A). Drug treatment led to marked increase in genomic binding in both p53$^{R158G}$ and p53$^{wt}$ cells, with higher signal intensity around the TSS-proximal regions (Fig. 5a, Supplementary Fig. 7A), suggestive of DNA-binding capability of both p53 isoforms. Importantly, predicted motif analysis for TSS-proximal peaks suggested that p53 consensus motifs were enriched not just in p53$^{wt}$ but also in p53$^{R158G}$ (motif similarity $P = 0.029$ and 0.039, TOMTOM match statistic; $P = 1 \times 10^{-16}$ and $1 \times 10^{-265}$, HOMER statistic) (Fig. 5a, Supplementary Fig. 7A). However, by comparing the binding patterns of wtp53 and mutp53 to TSS, we found that the mutp53$^{R158G}$ peaks were highly dissimilar from the peaks of wtp53 (Supplementary Fig. 8, column 1 row 3). In addition, we aligned the ChIP-seq data from the cells treated with the belinostat/cisplatin combination, and showed that TSS-proximal peaks post-treatment were distinct from the vehicle control in both wtp53 and mutp53. Interestingly, peaks enriched in drug-treated mutp53$^{R158G}$ cells resembled that of drug-treated wtp53 (Supplementary Fig. 8, column 2 row 4). These data collectively suggest that mutp53$^{R158G}$ partially retains wtp53 activity when cells are exposed to stress signal, which supports the induction of wtp53 response genes by the drug-induced transactivation of p53$^{R158G}$ in H2170 cells (Supplementary Fig. 4A), as well as the increased DNA-binding affinity of mutp53 upon acetylation (Supplementary Fig. 4B, C).

To identify the genes activated by acetylated p53$^{R158G}$ which are associated with the induction of apoptosis, we compared the treatment-induced perturbations of transcriptomic profiles (by Ampliseq RNA sequencing) between wild-type and mutant cells. These gene panels were subjected to KEGG ontogeny enrichment analysis. Expectedly, genes mediating p53 signaling were elevated in p53$^{wt}$ cells after treatment ($P$-value= $7.87 \times 10^{-11}$); in p53$^{R158G}$, genes associated with cell cycle, DNA replication and DNA damage repair pathways were prioritized (Supplementary Fig. 7B). We further analyzed gene sets obtained from both ChIP-seq and AmpliSeq for concordance of significantly perturbed genes pre- and post-treatment ($\log_2 > 1.5$), also found to be bound by p53 within the proximal regions (<10 kb from TSS), as direct p53 targets (Fig. 5b, Supplementary Table 4). Selected genes were validated with qPCR, and consistently, direct p53$^{wt}$ binding increased expression of canonical p53 downstream genes (*MDM2*, *GADD45A*, *PMAIP1*) (Supplementary Fig. 7C-E, I-K). Among the mutp53-bound target genes, *TRAIP* and *RAD51* were markedly induced ($\log_2$-fold = 2.72 and 2.767, respectively) after drug treatment relative to wild-type cells (Supplementary Table 4, Fig. 5c, d, Supplementary Fig. 7F, L). Work on other p53 hotspot mutants have described part of the mechanistic basis for mutp53's GOF as direct interaction with E26 transformation-specific (ETS) motifs[37]. Two chromatin regulatory genes downstream of ETS2, *KMT2D*, and *KAT6A*[22], reported to be positively associated with hotspot GOF mutp53 (R175H, R248Q, R248W, R249S, or R273H), were neither transactivated nor upregulated in p53$^{R158G}$ (Fig. 5f, g, Supplementary Fig. 7G, H), thereby distinguishing DNA binding by Arg$^{158}$ mutants from other mutated p53 GOF isoforms.

RAD51 and TRAIP are key components of the DNA damage and repair mechanisms[38–40]. However, combination-treated p53$^{R158G}$ cells did not demonstrate increased DNA damage response when compared to p53 loss or wtp53 cells, based on the analysis of induced-γH2AX (Fig. 5h), a surrogate marker for DNA damage, and the quantification of broken DNA measured using comet tail length (Supplementary Fig. 9A, B). Therefore, it is unlikely that the enhanced cytotoxicity in p53$^{R158G}$ is due to altered DNA damage signaling. We postulated that NFκB pathway regulation by TRAIP, a RING domain-containing E3 ubiquitin ligase[41], may explain the altered mutp53 function, as TRAIP is known to suppress NFκB[42–45]. We showed that TRAIP upregulation in p53$^{R158G}$ cells correlated with the IκB dephosphorylation and stabilization (Fig. 5h). Active IκB inhibits NFκB through blocking the nuclear import of p65[46]. Accordingly, we hypothesize that acetylation of p53$^{R158G}$ may impede p65 activation through perturbation of the TRAIP-IκB-NFκB axis. However, we noted in the qPCR and immunoblotting analyses

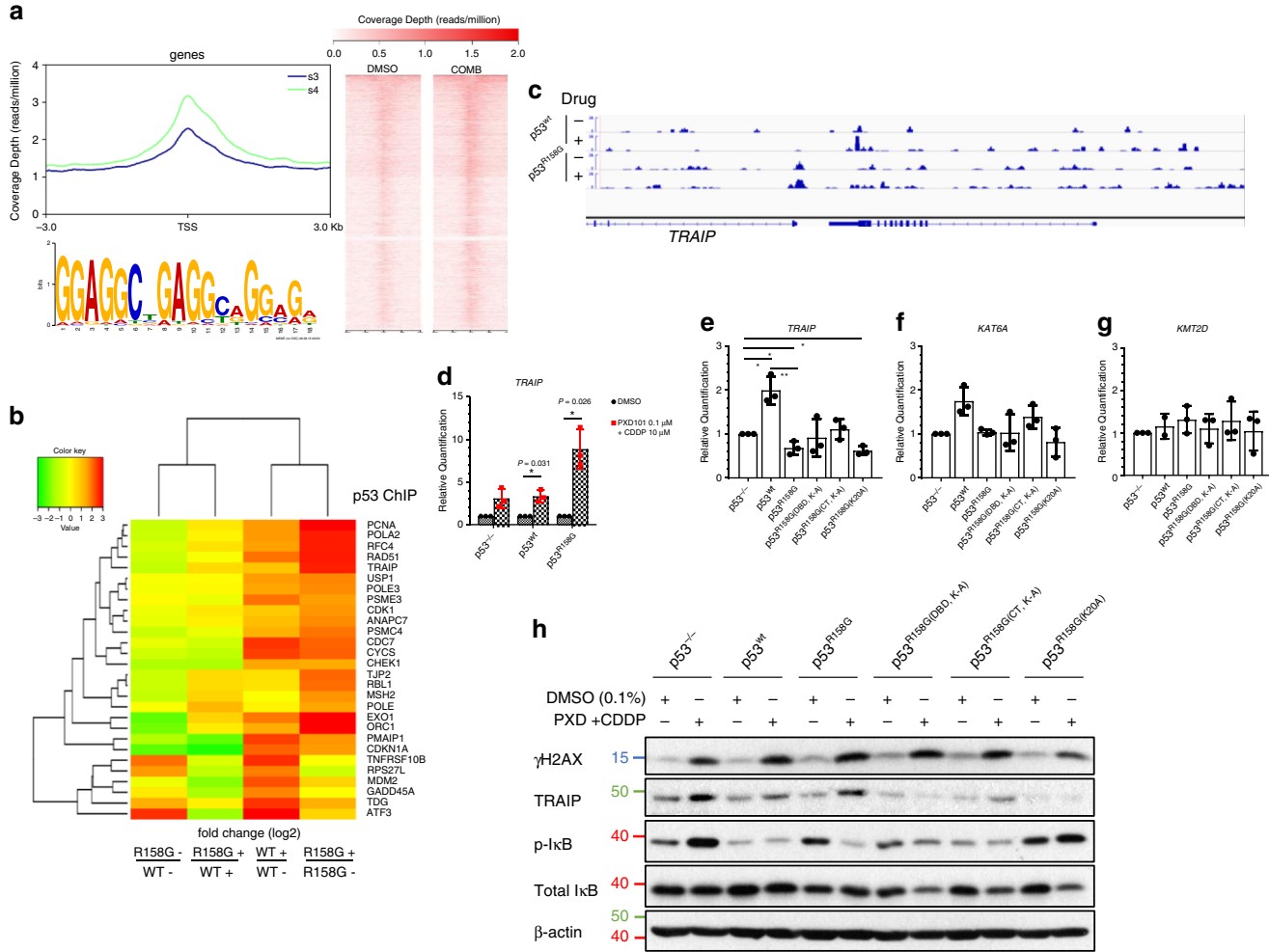

**Fig. 5 Genome-wide binding and downstream transactivation of GOF p53[R158G]. a** mutp53[R158G]-binding loci identified by p53 ChIP-seq analysis on Calu-1 cells treated with vehicle or belinostat/cisplatin combination for 24 h ($n = 1$). Meta-peak analysis showing distribution of p53[R158G]-binding sites across 3000 bp from the TSS of the nearest downstream gene (left, top). Density heatmap of the mutp53-binding sites (±3000 bp from TSS) examined by ChIP-seq (Right). Canonical wild-type p53 consensus motif was identified by MEME/TomTom from the TSS-proximal ChIP-Seq peaks (bottom left). **b** Comparisons of the transcriptomic levels of differentially-expressed genes (AmpliSeq analysis) bound by p53 (ChIP-seq analysis) under vehicle (−) or drug-treatment (+) conditions. Colors from green to red indicate increasing RNA level (fold change as indicated) in each pair. Individual fold change values were tabulated in Supplementary Table 4. **c** Integrative Genomics Viewer display of mutp53 occupancy over promoter region of *TRAIP* gene in vehicle- or drug-treated cells (p53[wt], p53[R158G]). **d** RT-qPCR quantification of *TRAIP* gene in Calu-1 (p53[−/−], p53[wt], p53[R158G]) cells 48 h after vehicle or drug treatment. **e–g** RT-qPCR analyses of the basal mRNA levels of *TRAIP* (**e**), *KAT6A* (**f**) and *KMT2D* (**g**) in various p53 clones, compared to p53[−/−]. Data are represented as average relative quantification ± SD ($n = 3$ independent experiments). Two tailed Student's *t*-test; *$P < 0.05$; **$P < 0.01$. **h** Immunoblotting was performed to evaluate the effects of belinostat/cisplatin on the indicated targets in various p53 clones. β-actin shown as loading control ($n = 3$ independent experiments).

that TRAIP was elevated to a certain extent in p53[wt] and p53[−/−] (Fig. 5d, h). As TRAIP is involved in cellular response to genotoxic lesions such as replication-associated DNA damage[39], we reasoned that TRAIP is induced as part of a p53-independent physiological response to cisplatin-induced DNA cross-linking, particularly in p53 null cells. To confirm this, we extended our analyses on the remaining Calu-1 clones expressing p53[R158G]. Belinostat/cisplatin co-treatment consistently elevate TRAIP and suppress IκB phosphorylation in these clones, whereas the same treatment condition did not alter IκB in p53[wt] and p53[−/−] cells (Supplementary Fig. 10A, B).

**Acetylated p53[R158G] suppresses NFκB signaling through TRAIP.** To prove this hypothesis, we first determined the changes in chromatin binding affinity of acetylated mutp53 in the

context of *TRAIP* transactivation. Using ChIP-quantitative PCR (ChIP-qPCR), we validated the increased binding of p53[R158G] after belinostat/cisplatin treatment to the promoter region of the *TRAIP* gene (Supplementary Fig. 11A–C). In parallel experiments involving p53[wt] and p53[R158G(K20A)] cells, there was no apparent enrichment in term of wtp53 binding to *TRAIP*, and more importantly, the ChIP-qPCR signal of mutp53 binding was attenuated in the p53[R158G(K20A)] variant, which cannot be acetylated (Supplementary Fig. 11B, C). Collectively, these findings strengthened the concept that acetylation of p53[R158G] leads to the transactivation of *TRAIP* (Fig. 5d).

We next sought to investigate the relevance of *TRAIP* transactivation on NFκB signaling in the context of differential p53 alterations. NFκB stimulation renders resistance to programmed cell death through nuclear translocation of its activated heterodimeric complex, DNA interaction and induction of

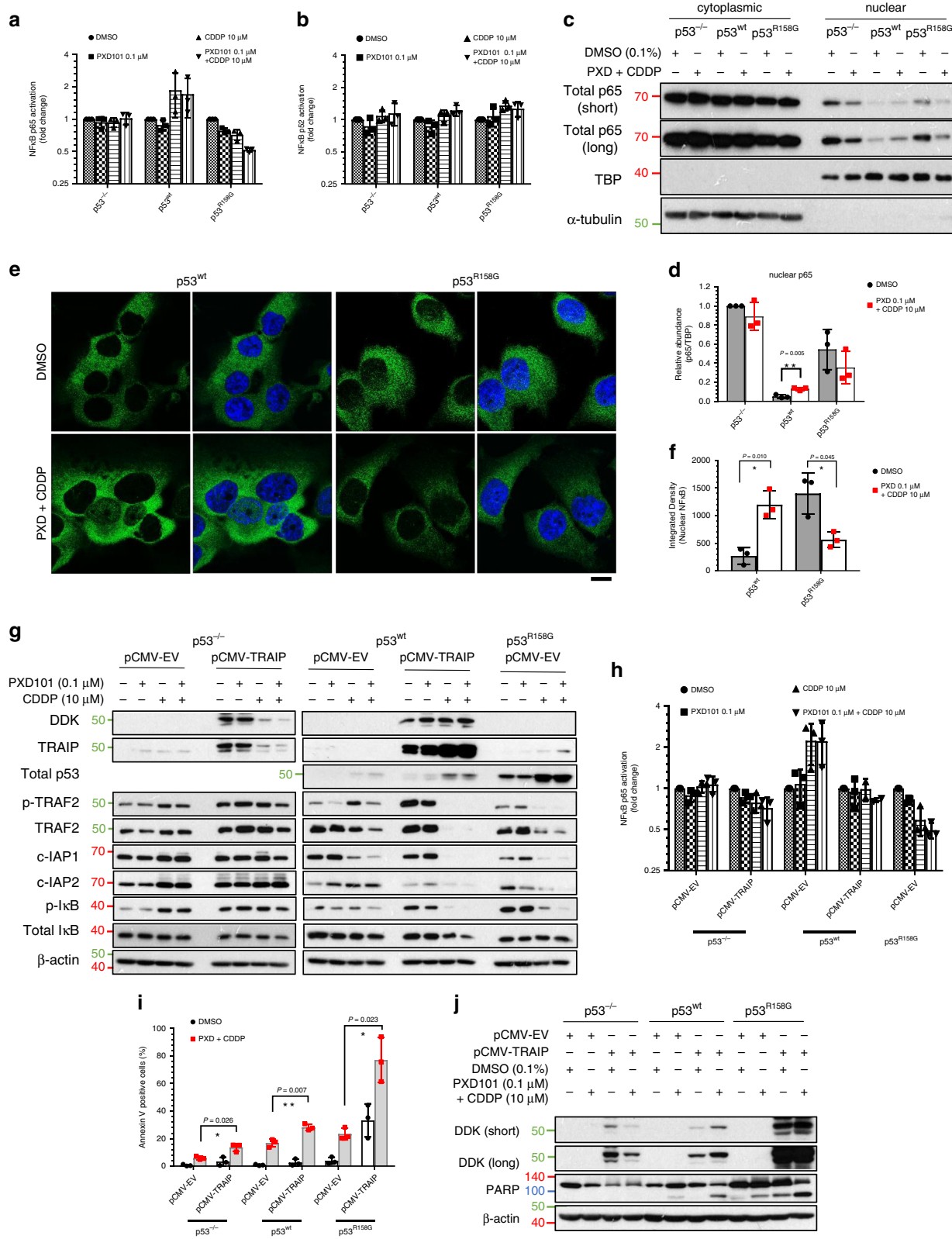

survival signals, which is regulated by an auto feedback loop involving its negative regulator IκB[47]. Activation of NFκB is determined either by the canonical NFκB1/RelA complex or the non-canonical NFκB2/RelB complex. While wtp53 and NFκB are often coordinated in an opposing manner due to their contrasting cellular functions[48,49], mutp53 proteins have been shown to promote NFκB activation[11]. In the context of LUSC, we first

showed using DNA-binding ELISAs that cisplatin treatment increased p65 (RelA, NFκB p65 subunit) transactivation in p53[wt] cells (Fig. 6a). In contrast, a reciprocal downregulation was observed in p53[R158G] cells, with a further reduction observed when combined with belinostat; whereas NFκB was unaffected in p53 deficient cells (Fig. 6a). To rule out the possible alternative involvement of non-canonical NFκB signaling, we investigated

**Fig. 6 Drug-induced TRAIP expression negatively regulates NFκB signaling in p53$^{R158G}$ cells. a, b** Quantification of active nuclear NFκB1 (p65) (**a**) and NFκB2 (p52) (**b**) in Calu-1 (p53$^{−/−}$, p53$^{wt}$, p53$^{R158G}$) cells 48 h after vehicle or drug treatment. Data are presented as mean ± SD ($n = 3$ independent experiments). **c, d** Distribution of p65 (in cytoplasm or nucleus) was determined by Western blot (**c**) after nuclear-cytosolic fractionation. TATA-box binding protein (TBP) (nuclear) and α-tubulin (cytoplasmic) were used as loading controls. Nuclear p65 expression was quantified with densitometry after normalizing to TBP (**d**). Data are presented as mean ± SD ($n = 3$ independent experiments). Two tailed Student's $t$-test; **$P < 0.01$. **e, f** Immunofluorescence staining was performed to determine the localization of p65 (Alexa Fluor-488) in p53$^{wt}$ and p53$^{R158G}$ cells 48 h post-treatment. At least five independent fields were taken for each condition with a minimum of 30 nuclei per group ($n = 3$ independent experiments). Representative confocal images shown at 63× magnification (**e**). Scale bar, 10 μm. Merged images displayed with blue indicates DAPI, green indicates p65. Integrated density quantifying nuclear NFκB (p65) in p53$^{wt}$ and p53$^{R158G}$ cells (**f**). Data are presented as mean ± SD ($n = 3$ independent experiments). Two tailed Student's $t$-test; *$P < 0.05$. **g** Western blot showing effects of TRAIP overexpression on the indicated proteins in Calu-1 isogenic cells (p53$^{−/−}$, p53$^{wt}$, p53$^{R158G}$) 48 h after vehicle or drug treatment. β-actin shown as loading control ($n = 3$ independent experiments). **h** Quantification of active nuclear NFκB1 (p65) in Calu-1 (p53$^{−/−}$, p53$^{wt}$, p53$^{R158G}$) cells after TRAIP overexpression with the indicated drug treatment. Data are presented as mean ± SD ($n = 3$ independent experiments). **i, j** Annexin V staining (**i**) and Western blotting (**j**) in TRAIP overexpressed cells 48 h post-treatment. Data are presented as mean ± SD ($n = 3$ independent experiment). Two tailed Student's $t$-test; *$P < 0.05$, **$P < 0.01$.

the transactivation of p52 (NFκB2 subunit) and showed that DNA-binding was marginally affected in all cell types (Fig. 6b)[50]. Cytoplasm to nuclear shuttling of p65 is critical for the initiation of the NFκB pathway[51]. In p53$^{R158G}$ cells, drug treatment reduced nuclear accumulation of p65; conversely, pronounced p65 nuclear translocation was observed in wild-type cells, in concordance with the increase in DNA-binding assays (Fig. 6c–f). The wtp53-p65 relationship was illustrated experimentally in colon cancer cells HCT116 p53 wild-type/null pairs[52], where drug combination induced p65 nuclear translocation in wild-type (HCT116$^{+/+}$) but not in null (HCT116$^{−/−}$) cells (Supplementary Fig. 12A–D). In addition, we observed that basal nuclear NFκB is kept at levels lower in p53$^{wt}$ than in p53$^{R158G}$ or p53 null cells (Fig. 6c–f, Supplementary Fig. 12A, B), likely through higher *TRAIP* expression in wtp53 cells (Fig. 5e). Together, our findings support the contention that increased transcription of *TRAIP* by acetylated p53$^{R158G}$ leads to inhibition of NFκB through blocking IκB phosphorylation and nuclear import of p65.

To elucidate the mechanisms relating TRAIP to NFκB inhibition, we overexpressed TRAIP in both p53$^{−/−}$ and p53$^{wt}$ cells. Surprisingly, cisplatin alone or in combination down-regulated exogenous TRAIP in p53$^{−/−}$ cells without affecting the downstream targets (Fig. 6g, left). In contrast, ectopic TRAIP expression was sustained in p53$^{wt}$ cells, leading to the down-regulation of TRAF2, c-IAP1, and c-IAP2, pro-survival components of the TNF receptor family[53], potentially through the ubiquitin-proteasomal degradation pathway (Fig. 6g, right). These data are consistent with previous work demonstrating that TRAF2 and c-IAP are downstream effectors of TRAIP[42,45]. As previously described[54], ablation of TRAF2 reduced phosphorylation and degradation of IκB (Fig. 6g, right), therefore facilitating IκB-mediated impediment of NFκB transactivation (Fig. 6h). This negative NFκB regulation by exogenous TRAIP sensitized p53-expressing cells to drug-induced apoptosis, as indicated by increased Annexin V positive population and PARP cleavage (Fig. 6i–j). Importantly, exogenous TRAIP recapitulated p53$^{R158G}$ response to drug treatment in p53$^{wt}$ cells. Consistently, acetylation of p53$^{R158G}$ is associated with p21 upregulation, p-IκB suppression and elevated *TRAIP* transcript in H2170 cells (Supplementary Fig. 13A–C). Of note, these modulations were consistent in other cell types expressing Arg$^{158}$ mutp53 (H441, H661), but were not found in other GOF mutp53 cells (H1417, H1975, HCC70, MDA-MD-468, and SK-BR-3 cells) (Supplementary Fig. 13A, B), thereby highlighting the mechanistic distinctiveness of the Arg$^{158}$ mutp53. Effective silencing of *TRAIP* markedly rescued cellular apoptosis in H2170 cells, as demonstrated by the reduced PARP and caspase 3 cleavage, by sustaining p-IκB without affecting p53 acetylation (Supplementary Fig. 13D). It is interesting to note that TRAIP expression was

consistently detected in the DMSO-treated cells upon siRNA interference, and we reasoned that the presence of a compensatory mechanism could potentially stabilize TRAIP and counterbalance the knockdown effect. We postulate that this mechanism could be impaired under cellular stress, thus allowing knockdown of TRAIP to be possible upon drug treatment. However, the detailed mechanism remains elusive at this moment.

To further investigate the mechanisms of the TRAIP/TRAF2/NFκB axis, dominant negative IκBα mutants (IκBα-ΔN, IκBα-S32A) were introduced to shut off NFκB activity. As expected, dephosphorylation of IκB in wtp53 cells reduced p-p65, p-TRAF2, and c-IAP2 upon drug treatment, recapitulating the effects of TRAIP overexpression (Supplementary Fig. 16A). Surprisingly, transfection of IκBα mutants has little effect in p53$^{−/−}$ cells, and conversely, p65 was highly phosphorylated. Functionally, NFκB inhibition also induced pronounced apoptosis in wtp53 and mutp53 cells, but to a lesser extent in null cells (Supplementary Fig. 16B). Collectively, these data strongly suggest a direct role of TRAIP, expressed upon acetylation of mutp53$^{R158G}$, in facilitating apoptosis through disruption of NFκB pathway signaling.

**Potent p53 acetylators reduce mutp53 tumor growth.** Further analyses of compounds found active against p53$^{R158G}$ cells revealed that eight out of the 10 compounds induced p53$^{R158G}$ acetylation (Supplementary Fig. 15A), which was correlated to increased p21, elevated *CDKN1A* and *TRAIP* transcripts, suppression of TRAF2 and p-IκB, as well as augmented PARP cleavage (Supplementary Fig. 15A, D, E), thus supporting our earlier observations. On the contrary, exposure to these acetylators induced stabilization of wtp53 and upregulated p21, increased *MDM2* and *CDKN1A* transcripts, but had minimal impact on p-IκB and *TRAIP* mRNA (Supplementary Fig. 15B–E), thereby highlighting the specificity of these single agents in targeting mutp53$^{R158G}$ cells.

When applied in vivo, single agent cisplatin was effective in reducing tumor growth of p53$^{R158G}$ xenograft as compared to vehicle, with no efficacy observed in p53$^{−/−}$ and p53$^{wt}$ tumors (Supplementary Fig. 16A–C). Of note, efficacy of cisplatin in the p53$^{R158G(K20A)}$ tumors with defective acetylating mechanism was evidently reduced (Supplementary Fig. 16A–C). Mechanistically, acetylation of mutp53$^{R158G}$ was accompanied by elevated TRAIP (Supplementary Fig. 16E), p65 inhibition (Supplementary Fig. 16F) and reduced Ki67 staining, a marker for cellular proliferation (Supplementary Fig. 16G); in contrast with increased p65 nuclear accumulation and lack of efficacy in the wtp53 tumors. Scoring of *TRAIP* transcripts demonstrated a reciprocal inverse correlation with NFκB activity, for instance, a high basal score was associated

with low p65 accumulation in p53$^{wt}$ tumors (Supplementary Fig. 16H). Further analyses of vehicle- and cisplatin-treated xenograft for drug-activity showed elevated *TRAIP* mRNA copies in p53$^{R158G}$ tumors, but decrease in wild-type tumors, validating the in vitro data (Supplementary Fig. 16H). We expanded our in vivo studies to screen for efficacy of other potent acetylating agents, as the clinical application of HDACi has been challenging in solid cancers, possibly due to their poor therapeutic index[55]. Whereas volasertib demonstrated high toxicity in mice (died within days of administration), tolerable doses of JQ1 and topotecan were effective in suppressing growth of p53$^{R158G}$ tumors compared to the p53$^{wt}$ and p53$^{-/-}$ counterparts (Supplementary Fig. 16I-K), providing direction for future development of p53-acetylating agents in p53$^{R158G}$-positive tumors.

Patient-derived xenograft (PDX) tumors of gastric cancer (GC; p53$^{R158C}$) and hepatocellular carcinoma (HCC; p53$^{R158H}$) harboring Arg$^{158}$ mutp53 were selected for evaluation with p53-acetylators: cisplatin, JQ1 and topotecan. All three compounds significantly reduced tumor growth in both PDX models (Fig. 7a, b, d, e) at tolerable doses (Fig. 7c, f). Tumor specimens from GC-PDX showed that cisplatin and topotecan significantly induced p53 acetylation and phosphorylation, with TRAIP upregulation as well as PARP cleavage observed in most of the treated tumors (Fig. 7g). Moreover, we demonstrated using RNA in situ hybridization the transcriptional induction of *TRAIP* mRNA in the drug-treated tumors (Fig. 7h, i), which was concordant with the protein expression in the matched tumor samples (Fig. 7g). These findings suggest that, while the Arg$^{158}$ p53 mutation is more prevalent in lung carcinomas, the mechanism conferring tumor cytotoxicity could be applied across cancer types harboring the same alteration. We further explored the potential of combining a DNA-damaging agent (cisplatin) with an effective p53-acetylating agent (topotecan) in both PDX models. Compared with single agents, combination of cisplatin/topotecan rapidly and profoundly suppressed tumor growth with no significant weight loss observed in all animals (Fig. 7j), with tumor regression observed in multiple cases (Fig. 7k).

## Discussion

Missense mutations in the DNA-binding domain constitute more than 70% of tumor-associated p53 mutations. These mutations alter DNA-binding capability of mutp53, concomitantly leading to a variable loss of p53 tumor suppressive functions, while mediating oncogenic GOF through transcriptional aberrations involving chromatin remodeling and interaction with transcription cofactors such as SREBP, ETS2, or NRF2[6,22,37,56]. Therapeutic interventions that have been proposed to target mutp53 cells include attempts to exploit the loss of checkpoint function[27,28], inhibit the GOF pathways[6,29], and direct destabilization of the mutated protein;[9,10] but these have yet to gain clinical traction. Whether these strategies can be applied globally to all mutp53 remains debatable, as only the six most common "hotspot" mutants have been studied in depth. We studied a GOF Arg$^{158}$ mutp53 in the DBD found to be relatively frequent in lung carcinomas, and elucidated a unique pathway of mutp53 that can be exploited therapeutically.

The complex crosstalk between p53 and NFκB has been described previously, with the suppression of p65 in wild-type cells a result of competitive interaction with p300 or glucocorticoid receptor;[48,49] conversely mutp53 prolongs and enhances NFκB activation[11,12,57,58]. We propose here a mechanism for this divergent effect through acetylation of Arg$^{158}$ mutp53, which alters its DNA-binding spectrum and upregulates *TRAIP* as key target gene, leading to NFκB suppression through TRAF2

degradation and culminating in cell death (Fig. 8). This p53 isoform appears to carry distinctive GOF mechanisms that does not involve the chromatin regulatory genes as observed in other DBD hotspot mutants; other hotspot p53 mutants, such as R175H and R273H, did not upregulate TRAIP in response to belinostat and cisplatin treatment. Given that a point mutation is sufficient to affect p53 transactivation[59], and that protein misfolding regulated by zinc-binding could affect p53 transcriptional functions[60], it is possible that Arg$^{158}$ mutp53 modulates site-specific DNA binding through aberrant Zinc2+ interaction which should be explored further. One possible caveat of the study is the lack of a universal cellular system to distinguish the functions of the various GOF mutp53. The inability to establish a robust CRISPR-Cas9 model has prevented us from studying this.

Our expression analysis and ChIP-Seq findings concordantly demonstrated the retention of transcriptional activity of mutp53$^{R158G}$, which has been described in other DBD variants. Exposure to cellular stress, such as cisplatin, could trigger a partial wild-type activity in mutp53$^{R158G}$ that induces p53-dependent pro-apoptotic signals, which is not seen in p53 null cells. However, the DNA-binding activity of mutp53$^{R158G}$ is distinct from p53$^{wt}$, manifesting in GOF activity. Paradoxically, this DNA-binding ability is prerequisite for the induction of apoptosis as initiated through mutp53$^{R158G}$ acetylation, a key post-translational event closely associated with its transcriptional activity and stability[61,62]. Loss of acetylation, particularly at the DBD, significantly impacted cell cytotoxicity, indicating that this post-translational event is indispensable for mutp53 activation. We studied this using in silico modeling of the expected molecular disruptions effected by the acetylation of DBD, and our simulation suggested strongly that the distorted mutp53 conformation could be restored through acetylation within the DBD, which appears to act as a stabilizing factor to facilitate the formation of a functional and stable mutp53 dimers with strong DNA-binding affinity. These further refined the commonly held credence that targeting acetylation and eventual depletion of mutp53 is a plausible anti-tumor approach[63]. Such findings are made therapeutically relevant by positive screen hits for compounds with pharmacological activity of p53$^{R158G}$ acetylation. The selective cytotoxicity of these compounds in Arg$^{158}$ mutp53 tumors, compared to wild-type and null tumors, is promising as a companion biomarker for selecting patients with greater potential to respond to the p53-acetylating agents. From the therapeutic standpoint, dosages of acetylating agents used in the xenograft experiments were well-tolerated, further heightening the possibility of clinical translation. For instance, the bromodomain (BRD) inhibitor JQ1 is currently in clinical development to target a rare genomic BRD-NUT fusion event in NUT midline carcinoma[64], and our data extends potential evaluation in Arg$^{158}$ mutp53 tumors. Curated data from public databases (IARC TP53, COSMICs, ICGC, and cBioportal) reveal the presence of Arg$^{158}$ mutp53 in multiple carcinomas; and the effectiveness of mutp53 acetylators in the PDX models of gastric and liver cancers further extends the applicability of these findings to other cancer types.

In conclusion, we provide evidence of a unique mechanism of p53 activation that is specific to Arg$^{158}$ mutations, exposing a previously unrecognized therapeutic vulnerability and facilitating a biomarker-implemented approach directed against this GOF mutant. This is unsurprising, as it has been recently recognized that different p53 mutations are attributable to distinctive phenotypes and cellular effects[65,66], and our work adds to this. If validated in the clinic, this represents an advance in therapeutic treatment of tumors with mutated p53, particularly that of LUSC, in which Arg$^{158}$ mutant is found to be prevalent.

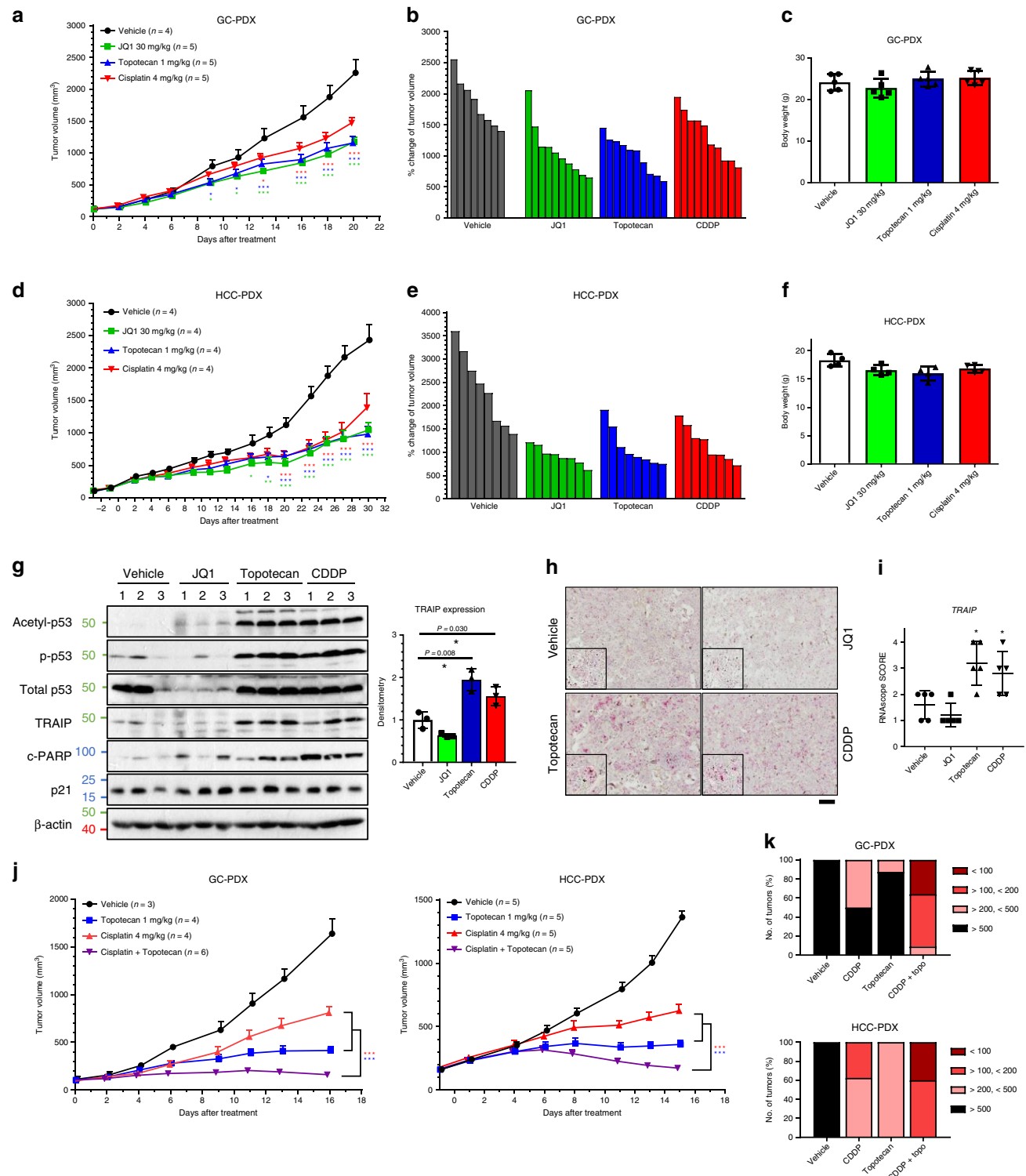

## Methods

**Genomic analyses of human cancers**. The pan-cancer mutation annotation file were downloaded from GDAC (version 2016_01_28[25]). Mutations found on *TP53* genes and the corresponding protein change were extracted for analysis.

**Cell culture**. Cancer cell-lines (Calu-1, ChaGo-k-1, HCC70, H441, H520, H596, H661, H747, H1417, H2170, MDA-MB-468, SK-BR-3, SK-MES-1) and lung fibroblast cells (MRC-5) were obtained directly from the American Type Culture Collection (ATCC), while others (A549, BT-549, H1975, MIA-Paca-2, PANC-1) were provided by Ashok Venkitaraman (MRC Cancer Unit, Cambridge, UK). Cultures were maintained in a humidified 37 °C incubator, in culture medium (according to ATCC's recommendation) supplemented with 10% fetal bovine

serum, 2 mM L-glutamine, 100 μg mL⁻¹ streptomycin and 100 U mL⁻¹ penicillin. Paired HCT116 p53⁻/⁻ and p53⁺/⁺ cell-lines were obtained from Bert Vogelstein (The John Hopkins Medical School, Baltimore, MD) as a gift. Isogenic Calu-1 cells stably expressing pcDNA3.1 vector, pcDNA3.1-p53wt, pcDNA3.1-p53R158G, pcDNA3.1-p53R158G(DBD, K-A), pcDNA3.1-p53R158G(CT, K-A), pcDNA3.1-p53R158G(K20A) plasmids, respectively, were generated by transfection and single-cell selection.

**Transwell invasion assay**. Matrigel invasion assay was carried out with Corning® BioCoat™ Matrigel Invasion Chambers (#354480). In brief, 25,000 (for Calu-1) or 50,000 (H2170) cells were plated in the transwell chamber with serum-free medium. For gene silencing experiments, cells were incubated with siRNA overnight

**Fig. 7 Cisplatin, JQ1 and topotecan reduce tumor growth in PDX models harboring Arg[158] p53 mutations. a–f** Efficacy of cisplatin, JQ1 and topotecan were investigated in PDX model of gastric cancer (GC) and hepatocellular (HCC) cancer. Growth curve analysis of GC-PDX (**a**) and HCC-PDX (**d**) treated with vehicle, cisplatin (CDDP; 4 mg kg[−1]), JQ1 (30 mg kg[−1]), or topotecan (1 mg kg[−1]). Tumor sizes are presented as mean ± SEM. Two-way ANOVA with Bonferroni correction; $*P < 0.05$; $**P < 0.01$; $***P < 0.001$. Waterfall plots showing change in tumor volume (relative to initial tumor volume at treatment-start day) for each individual GC-PDX (**b**) and HCC-PDX (**e**) tumor in respective treatment group. Bodyweight of mice with GC-PDX (**c**) and HCC-PDX (**f**) at assay endpoint was tabulated as mean ± SD. **g** Western blots demonstrating changes of the indicated proteins in tumors of respective treatment ($n = 3$ independent tumors). β-actin shown as loading control. Densitometric quantification of TRAIP expression was tabulated on the right. Relative fold change is normalized to β-actin, relative to vehicle control tumors. Data presented as mean ± SD. Two tailed Student's $t$-test; $*P < 0.05$, $**P < 0.01$. **h–i** RNA in situ hybridization (RNAscope) showing *TRAIP* expression in respective tumors. Representative images showing *TRAIP* mRNA signal are shown at ×20 magnification (**h**). Scale bar, 50 µm. Semi-quantitative scoring (0–4) of *TRAIP* mRNA signal (dots/cell) in p53[wt] and p53[R158G], respectively, was tabulated (**i**). Data are represented as scattered dot plot ± SD ($n = 5$ independent tumors). Two tailed Student's $t$-test, $*P < 0.05$. **j–k** Efficacy of cisplatin and topotecan combination were investigated in PDX models of gastric cancer and hepatocellular cancer. **j** Growth curve analyses of GC-PDX and HCC-PDX treated with vehicle, cisplatin (CDDP; 4 mg kg[−1]), topotecan (1 mg kg[−1]) or CDDP/topotecan combination. Tumor sizes are presented as mean ± SEM. Two-way ANOVA with Bonferroni correction; $*P < 0.05$; $**P < 0.01$; $***P < 0.001$. **k** Change in tumor volume (relative to initial tumor volume at treatment-start day) for each individual tumor in respective treatment group were calculated and tabulated into four groups (>500%; >200% <500%; >100% <200%; <100%). Tumor volume below 100% indicates tumor regression.

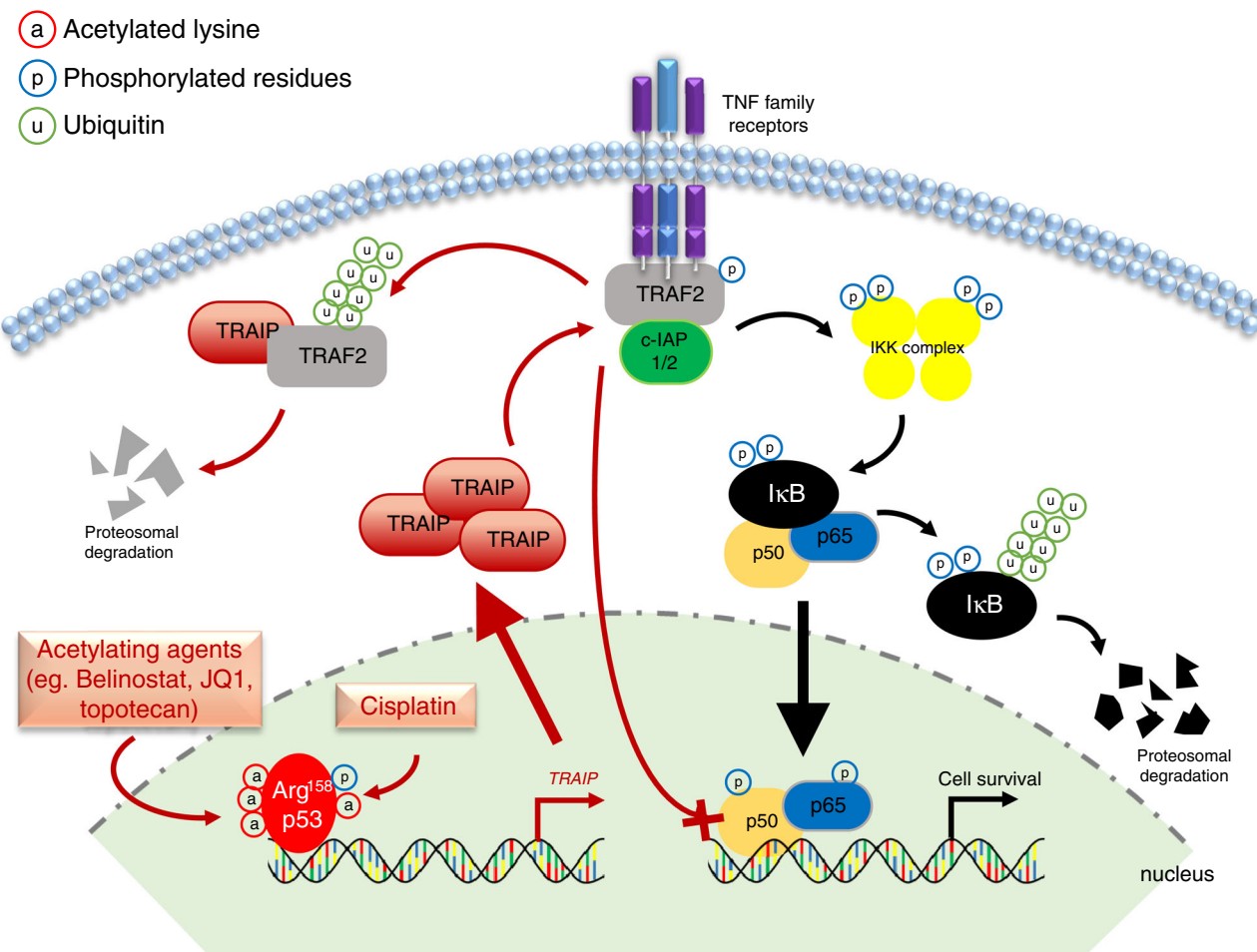

**Fig. 8 Proposed mechanism of Arg[158] mutp53 in response to treatment with acetylating agents.** Tumor cells harboring mutp53 expressed inherently high level of NFκB activity to confer cellular survival. We showed that in mutp53[R158G] cells, NFκB/p65 signaling could be suppressed through negative regulation of the TRAF2/c-IAP/IκB-axis via the upregulation of TRAIP, which is induced transcriptionally by monotherapy or combination treatment of compounds shown to induce mutp53 acetylation.

prior to plating. Tumor cells were allowed to invade for 24–36 hours before the transwell inserts were fixed with 4% paraformaldehyde, stained with gentian violet and imaged for analysis. Number of invaded cells was quantified with ImageJ Software (NIH), and expressed as mean ± SD.

**Anchorage-independent soft agar assay.** The colony forming assay comprises three layers of soft agar/medium matrix in 24-well plate: a bottom layer of 0.6% agar matrix; a middle layer containing 0.36% agar matrix with the indicated cells

(10,000 cells/well); and a top layer comprising complete medium. Culture medium were changed weekly. For gene silencing experiments, cells were incubated with siRNA overnight prior to culturing in soft agar. After 4 weeks of culture, colonies were stained with MTT stain (Promega, #G4000) for 2 h and imaged. Colonies were counted with ImageJ Software, and expressed as mean ± SD.

**PCR and sequencing.** Total RNA was isolated from cells using RNeasy kit (Qiagen, #74106) and reverse-transcribed to cDNA using iScript RT Supermix

(Bio-Rad, #1708841). Sanger sequencing were carried out on amplified DNA to sequence region of interest according to standard protocol. Real-time qPCR was performed in triplicates using standard procedures on a 7500HT Fast-Real-Time PCR platform (ABI), using probe-based assays. Gene expressions were normalized to the housekeeping gene *GAPDH*. ChIP was performed using Active Motif ChIP-IT ChIP-seq kit according to manufacturer's protocol with slight modifications. In brief, cells were crosslinked with 1% formaldehyde, glycine quenched, lysed with cell lysis buffer (Sigma, #2978) supplemented with 1 mM PMSF, and subjected to pulse sonication (20 s with 30 s internal, five cycles). The supernatant was then immunoprecipitated with the p53 and rabbit IgG antibodies (Cell Signaling, #9282 and #2729) at 4 °C overnight and incubated with Magna ChIP Protein G magnetic beads (Millipore, #16-662). The beads were washed before DNA was reverse-crosslinked and purified. For ChIP-Seq, sequencing libraries were prepared using TruSeq DNA LT Sample Prep kits, and pair-end sequencing was performed on Illumina Hi-Seq platform. ChIP-qPCR validation was performed in triplicates using standard procedures on a 7500HT Fast-Real-Time PCR platform (ABI), using primer-based assays. For Ion AmpliSeq Transcriptome Human Gene Expression (Thermo Fisher), 10 ng RNA was used for barcoded cDNA library preparation using PCR-based targeted-amplicon amplification according to man-ufacturer's protocol. Pooled libraries were amplified with emulsion PCR on Ion Torrent OneTouch2 instruments (OT2) and sequenced on Ion Torrent Proton sequencing system. Both Hi-seq and AmpliSeq were performed at AITbiotech (Singapore), Sanger sequencing at Axil Scientific (Singapore). All primer sequences used for RT-qPCR and ChIP-qPCR in this study are tabulated in Supplementary Table 5.

**Chemical screening, compounds, cell viability assay**. Selleckchem Anti-cancer compound Library (414 compounds, #L3000), Enzo Epigenetic Library (43 mod-ulators, #BML-2836) and Selleckchem Epigenetic Library (151 modulators, #L1900) were used for high-throughput chemical screening. Bleomycin (S1214), AZD7762 (S1532), cladribine (S1199), topotecan (S1231), nocodazole (S2775), volasertib (S2235), BI-2536 (S1109), 17-AAG (S1141), belinostat (PXD101, S1085), bosutinib (S1014), MK-5108 (S2770), JQ1 (S7110), CX-6258 (S7041), alisertib (S1133), JNJ-26481585 (S1096), and M344 (S2779) were purchased from Sell-eckchem; oxamflatin was purchased from Sigma Aldrich (O3139). All compounds were dissolved in Dimethyl sulfoxide (DMSO, Sigma Aldrich, #D8418). Cisplatin was obtained from Hospira Pharmaceuticals. Approximately 3000/1000 cells were seeded per well into a 96/384-well plate 24 h prior to drug treatment. For high-throughput studies, cultures were dispensed (50 μL) into 384-well plates using a MultiFlo dispenser, while drugs were 100-fold diluted in DMSO and kept at 1% (v/v) across all drug concentrations and control. The cell viability was assayed at 72 h post-treatment using either CellTiter-Glo luminescent or CellTiter-96 Aqueous One cell viability reagent (Promega, #G7572 and G3582). The luminescence signals were detected using TECAN Infinite M1000 pro multi-mode plate reader using an integration time of 1000/250 ms. Absorbance was recorded at 490 nm. The relative luminescence/absorbance units from treated wells were normalized against DMSO control wells and expressed as percentage cell viability. $IC_{50}$ values were calculated using GraphPad Prism software.

**Drug screening bioinformatics, Bliss Independent index**. Drug response of the Calu-1 cell-lines were quantified using the Inhibition score (*I*-score). For each compound *c*, the *I*-score was determined as follows:

$$I-\text{score}(c) = \left(1 - \frac{\text{Luminescence signal}(c)}{\text{Median of Luminescence signal}(\text{DMSO})}\right) \times 100$$

where the denominator is the median of the luminescence emitted in presence of Dimethy sulfoxide (DMSO). The inhibition heatmap of selected drugs (filtered at *I*-score >50 in at least one of the cell-lines) between the Calu-1 wild-type, mutant and null cell-lines were hierarchically clustered using complete linkage clustering and Euclidean distance measure. Combination Index (CI) was calculated based on the Bliss Independence model, by considering all dose combinations tested (observed combination response against the predicted combination response). By assuming that there is not drug-drug interaction, the combination effect is declared synergistic if the observed response is greater than the predicted response (Bliss < 0).

**Western blotting, cellular fractionation, and antibodies**. The sodium dodecyl sulfate–PAGE (SDS-PAGE) was performed as described previously[67]. Whole-cell lysates were extracted with cell lysis buffer (Sigma, #2978) supplemented with protease and phosphatase inhibitors (ThermoFisher, #78440). Protein concentra-tion was determined by BCA protein assay (Life Technologies, #23227), and equal amount of proteins were separated using SDS-PAGE. Cytosolic and nuclear frac-tionations were performed according to manufacturer's protocol (Active Motif, #40010). The ratio of cytoplasmic to nuclear proteins was 3:1 for Western blotting.
Antibodies used for immunoblotting include: PARP (#9542), cleaved PARP (#5625), caspase 3 (#9662), caspase 7 (#9492), acetyl-H3K23 (#9674), acetyl-H3K9K14 (#9677), total H3 (#9715), phospho-p53 Ser15 (#9286), acetyl-p53 Lys 382 (#2525), acetyl-p53 Lys379 (#2570), total p53 (#9282), p21 (#2947), γH2AX (#9718), phopho-IκB Ser32 (#2859), total IκB (#9247), total p65 (#8242), phoshph-TRAF2 Ser11 (#13908), total TRAF2 (#4724), c-IAP1 (#7065), c-IAP2 (#3130),

α-tubulin (#2125), and horseradish peroxidase (HRP)-conjugated β-actin (#5125) antibodies as well as secondary anti-mouse (#7076) and –rabbit (#7074) HRP-conjugated secondary antibodies were obtained from Cell Signaling Technologies; acetyl-H4 (#06-866) antibody was from Merck Millipore; TRAIP (#ab80170), TATA-box binding protein (#ab818) were from Abcam; DDK (#TA180144) antibody was from Origene. All antibodies were used at 1:2000 dilution.

**Caspase activity, apoptosis and cell cycle assays**. Caspase 3/7 enzyme activities were measured by using Caspase-Glo 3/7 Assay Systems (Promega, #G8090) according to manufacturer's protocol. Cells were incubated with indicated com-pounds for 24 h, and the luciferase activities were measured after substrate incu-bation with a TECAN Infinite plate reader. Apoptosis assay was carried out using annexin V-APC (Thermo Fisher, #A35110) and propidium iodide (PI, BD Bios-ciences, #556463) according to manufacturer's protocol. In brief, harvested cells ($0.2 \times 10^6$) were washed with PBS followed by annexin V buffer (Thermo Fisher, #V13246), and stained with Annexin V/PI mixture for 15 min at 25 °C. Cell cycle fractions were determined through PI nuclear staining. In brief, harvested cells ($0.2 \times 10^6$) were fixed with 70% cold ethanol, and stained with PI staining buffer (0.03 mg ml$^{-1}$ PI, 0.1 mg ml$^{-1}$ RNAse A, 0.1% Triton-X in PBS) for 30 min at 25 °C. The staining profile was acquired (≥10,000 cell events) using LSR II Flow Cytometer (BD Biosciences). The acquisition was analyzed using the FlowJo 8.1.1 software (Tree Star).

**Plasmids and transfection**. For stable overexpression in Calu-1 cells, wtp53 (pcDNA3.1-p53wt) and mutp53 (pcDNA3.1-p53R158G) sequences were cloned from MRC-5 and H2170 cell-lines, respectively, while site-directed mutagenesis was performed for substitution of Lysine residues to Alanine in the indicated plasmids [pcDNA3.1-p53R158G(DBD, K-A), pcDNA3.1-p53R158G(CT, K-A), pcDNA3.1-p53R158G(K20A)] using QuikChange Lightning Kit (Agilent, #210519). For stable knockdown of p53, H2170 cells were transfected with pLKO.1-puro Non-Target shRNA Control (shNT, Sigma Aldrich, SHC016), pLKO.1-puro Luciferase shRNA Control (shLuc, Sigma Aldrich, #SHC007), or shRNAs targeting *TP53* (gifts from Cheok CF, IMCB, A*Star, Singapore). For transient overexpression, pCMV6-Entry vector and pCMV6-TRAIP-DDK was obtained from Origene, while pCMV-IκB(ΔN) and pCMV-IκB(S32A) are gifts from Pieter Johan Adam Eichhorn (Cancer Science Institute of Singapore). Plasmid transfections were carried out with FuGENE HD (Promega, #E2311). For transient gene knockdown, *TP53* siRNA (sequence: 5'-ACUCCACACGCAAAUUUCCTT-3'), *TRAIP* siRNA (sequence: 5'-UUACACCUCAGGCUGGUCCCG-3') and All-Star negative control siRNA were obtained from Qiagen. siRNA transfection was conducted with JetPRIME reagent (Polyplus Transfection).

**Reporter-luciferase assay**. The transcriptional activity of p53 was assessed by reporter-luciferase assay. Cells were co-transfected with a combination of Cignal reporter plasmids (Qiagen, # CCS-004L), including a mixture of inducible p53-responsive firefly luciferase construct or non-inducible firefly luciferase construct, with constitutively expressing *Renilla* luciferase construct. The cells were then drug treated for the indicated durations, and luciferase assay was performed with Dual Luciferase Reporter Assay System (Promega, #E1910) according to manufacturer's instructions and detected by Lumat LB 9507 (Berthold Technologies). The ratio between *Renilla* luciferase across samples was used to normalize the transfection efficiency.

**In vitro translation (IVT), acetylation, and EMSA**. IVT was performed according to manufacturer's protocol (Thermo Scientific, #88881). Briefly, 1 μg each of pcDNA3.1-p53wt and pcDNA3.1-p53R158G plasmids were incubated with the Hela extract and reaction mix for 6 h at 30 °C. The in vitro protein acetylation (IVA) assay was performed, with a standard reaction included half of the sample from the IVT assay, 0.5 μg of histone acetyltransferase domain of p300/CBP pro-tein (Upstate, #14-418), 4 μg of Acetyl-CoA (Sigma, #A2056) and diluted in assay buffer (final concentration of 50 mM Tris–HCl, pH 8.0, 10% glycerol, 0.1 mM EDTA, 1 mM dithiothreitol). The reaction mix was incubated for 6 h at 30 °C in a shaking incubator. The expression and acetylation of the respective p53 proteins were validated with immunoblotting.
Electrophoretic mobility shift assay (EMSA) was conducted according to manufacturer's protocol (Thermo Scientific, E33075). 31-mer oligo containing p53 response element of *CDKN1A* gene (5'-CAGCATGCTCCAGGTAGAAGGAAA CAGGCCC-3') was ordered from IDT. Briefly, 50 ng of the synthesized oligos was incubated with 2 μl of the reaction containing the respective p53 isoforms, and incubated for 2 h at 30 °C. The samples were then separated by PAGE using 6% non-denaturing gel. DNA and protein were stained simultaneously with SYBR® Green nucleic acid gel stain and SYPRO® Ruby protein gel stain, respectively. The images were then visualized and captured using UV transilluminator.

**Molecular dynamics simulations of p53**. The dimeric structure of the p53 DNA-Binding Domain (DBD) complexed to a small fragment of DNA (PDB ID: 2AHI) was taken from the RCSB Protein Data Bank. Molecular dynamics (MD) simula-tions of the p53-DBD$^{WT}$, p53-DBD$^{R158G}$, and acetylated Lysine p53-DBD$^{R158G}$ were carried out with the pemed.CUDA module of the program Amber16 with the

Amber 14SB force field (ff14SB). Force field parameters for acetylated Lysine were taken as described previously[68]. The Xleap module was used to prepare the system for the MD simulations. All the simulation systems were neutralized with appropriate numbers of counterions. Each neutralized system was solvated in an octahedral box with TIP3P water molecules, leaving at least 10 Å between the solute atoms and the borders of the box. All MD simulations were carried out in explicit solvent at 300 K. During the simulations, the long-range electrostatic interactions were treated with the particle mesh Ewald method using a real space cutoff distance of 9 Å. The Settle algorithm was used to constrain bond vibrations involving hydrogen atoms, which allowed a time step of 2 fs during the simulations. Solvent molecules and counterions were initially relaxed using energy minimization with restraints on the protein and inhibitor atoms. This was followed by unrestrained energy minimization to remove any steric clashes. Subsequently the system was gradually heated from 0 to 300 K using MD simulations with positional restraints (force constant: 50 kcal mol$^{-1}$ Å$^{-2}$) on the protein over a period of 0.25 ns allowing water molecules and ions to move freely. During an additional 0.25 ns, the positional restraints were gradually reduced followed by a 2 ns unrestrained MD simulation to equilibrate all the atoms. Production simulations were carried out for 250 ns each in triplicates. Enhanced conformational sampling was carried out by subjecting the systems to accelerated MD (aMD) implemented in AMBER 16, using the "dual-boost" version. Conventional MD simulations carried out earlier were used to derive the aMD parameters (EthreshP, alphaP, EthreshD, alphaD). aMD simulations were carried out for 250 ns each. Simulation trajectories were visualized using VMD and figures were generated using Pymol.

**Chromatin immunoprecipitation (ChIP)-seq, AmpliSeq analyses.** At least 16 million 51-bp long reads were mapped to hg19 using bowtie v2.1.0 with parameters -N 1 –sensitive -p 2 –no-unal. Peaks were identified by MACS 2.0.9 using a maximum of 2 reads per unique position and otherwise default parameters. Only highly enriched (enrichment over background ≥5-fold, pileup ≥25) and highly significant (q-value < 0.01) peaks were shortlisted from the analysis. de novo motif search was conducted by MEME-ChIP54 using the TSS proximal peaks center extended by 75 bp in both directions to determine the distribution of wtp53 and mutp53-binding sites post-treatment with reference to hg19 annotations. MEME was programmed to search for the top 10 motifs with significant matches between query and target database using HOMER (Hypergeometric Optimization of Motif EnRichment) 55 v4.5. The heatmaps of p53 enrichments (wtp53 and mutp53$^{R158G}$) across 3 kb region from peak centre were plotted using ngsplot, v2.61. For AmpliSeq, fastq files were analyzed with the ampliSeqRNA plugin available for Ion Torrent sequencing platforms that used the Torrent Mapping Alignment Program (TMAP). TMAP first identify a list of Candidate Mapping Locations (CMLs) and aligned with Smith Waterman algorithm for specific and sensitive mapping. DEG analysis was performed using R/Bioconductor package DESeq2, with read count normalized to produce differential gene expressions as determined by p-value and the log$_2$ fold change. GEO superSeries: GSE129027.

**Comet assay.** Comet assay was performed using OxiSelect 96-well kit (Cell Biolabs, #STA-355) according to manufacturer's recommendation. Briefly, treated cells were trypsinized and mixed with low melting-point agarose at 37 °C. Cells/agarose mixture was quickly placed onto pre-coated 96-well Comet slide, and cooled at 4 °C to solidify the agarose layer. The ready slide was treated with hypertonic lysis buffer, and followed by incubation with alkaline buffer to unwind DNA strands and expose the alkali labile sites (comet tails), which were imaged by fluorescence microscopy at ×20 magnification (Olympus IX71/DP71). Olive Tail Moment (OTM) was quantified using ImageJ software (NIH, 1.51n) with OpenComet plugin, using the formula OTM = (mean Tail-Intensity – mean Head-Intensity) X Tail%DNA/100. The mean ± S.D. of OTM from at least 50 cells in each treatment group was quantified.

**Nuclear NFκB activity assay.** Activation of nuclear NFκB p65 subunit was quantified in nuclear protein extracts using the ELISA-based TransAM p65 NF-kB Kit (Active Motif, #40596) according to manufacturer's recommendation. Cellular fractionation was performed as described, and absorbance measurements (450 nm) on nuclear extracts (5 μg) in triplicates were performed using TECAN Infinite plate reader.

**High-content microscopy and immunofluorescence imaging.** Cultured cells were fixed with methanol for 30 min, and immunostained with acetyl-p53 (1:500, Cell Signaling, #2570), p-p53 (1:500, Cell Signaling, #9286), or NF-κB p65 (1:1000, Cell Signaling, #8242) for 4 hours at 4 °C, followed by incubation with Alexa Fluor-488 or −594 secondary antibodies (1:1000, Molecular Probes, Life Technologies, A11034 and R37121) for 2 h at 25 °C, and by Hoechst 33342 nuclear staining (1:2000, Sigma, #23491-52-3) for 15 min. Imaging of 96-well plate was performed using a high-content imaging system (MetaXpress Ultra, Molecular Devices) with ×40 objective. Airyscan super-resolution confocal images were acquired with LSM800 (Zeiss Germany) with ×63 objective. Image analyses were conducted using the Color Deconvolution Plugin in ImageJ software (NIH; 1.51n)[69].

**Tumor xenograft assay.** All in vivo xenograft studies adhered to the SingHealth Animal Care and Use Committee (IACUC) guidelines on animal use and handling. Usage of patient-derived materials have been approved by SingHealth Institutional Review Board (IRB). All tumor xenografts were established in male C.B-17 SCID mice aged 9-10 weeks and weighed 23–25 g (InVivos Pte. Ltd., Singapore). Mice were provided with sterilized food and water ad libitum, and housed in negative pressure isolators, which were set at 23 °C and 43% humidity, with 12-h light/dark cycles.

In brief, $10 \times 10^6$ isogenic Calu-1 cells (empty vector control, wtp53, mutp53) in 100 μL of PBS or 1–2 mm$^3$ dices of PDX tumors (GC-PDX and HCC-PDX) were injected subcutaneously into the flanks of each mice. Tumor size was monitored and measured three times weekly with Vernier calipers (tumor volume = length × width$^2$ × 3.14159/6). For inhibitor studies, mice were assigned into stratified groups based on average tumor volume (n = 5 animals/10 tumors per group). Treatment began when the tumor sizes reached ~200 mm$^3$. Cisplatin was provided in 0.9% NaCl and given at 4 mg kg$^{-1}$ by intraperitoneal (ip), once weekly for 3 weeks. JQ1 and topotecan were formulated in 30% (w/v) Captisol and 30 % (w/v) PEG300 at 30 mg kg$^{-1}$ and 1 mg kg$^{-1}$, respectively, and given via PO (orally), 7 days a week. Bodyweight at sacrifice and tumor samples were collected 3 weeks after treatment commenced.

**Immunohistochemistry (IHC), RNA in situ hybridization (ISH).** Xenograft tumors from control and compounds administered mice were fixed in formalin after surgical excision overnight, and embedded in paraffin and sectioned (4 μm). Tumor sections were deparaffinized using standard histologic procedures prior to antigen retrieval. For IHC, slides were incubated with primary antibodies [NF-κB p65 (1:1000, Cell Signaling, #8242), cleaved caspase 3 (1:1000, Cell Signaling, #9661), Ki67 (1:1000, Abcam, #ab15580)] overnight at 4 °C. Color development was performed using EnVision+ System-HRP (DAB) kit (Dako, Agilent Technologies) according to the manufacturer's protocol. Single molecule mRNA ISH (RNAscope, ACD Biotech) was performed using the Advanced Cell Diagnostics RNAscope 2.5 HD Detection kit (#322360) according to manufacturer's recommendation. TRAIP probe was customized. For both IHC and RNAscope, cell nuclei were counterstained with Hematoxylin (Thermo Scientific, #72604). Slide images were captured under optical imaging microscope at ×20 magnification (Olympus IX71/DP71). Images were acquired from five tumor samples each for every treatment group, and analyzed with ImageJ software using an image processing workflow described previously[67]. Briefly, Hematoxylin and DAB/Fast Red color-separated images were derived from the original image using ImageJ Color Deconvolution plugin. Object segmentation using the Particle Analyzer plugin was then performed to identify the individual cellular regions in each image, and the signal intensity (DAB) or count (Fast Red) in the image was then tabulated. IHC staining was analyzed as the percentage of DAB-positive cells/100 cells in each slide image. RNA-ISH staining was scored based on the number of positive signal in each cell (0: no staining or <1 dot/ 10 cells; 1: 1–3 dots/cell; 2: 4–9 dots/cell; 3: 10–15 dots/cell; 4: >15 dots/cell). At least 20 images were analyzed for each tumor sample.

**Statistics, data reporting, and reproducibility.** No statistical method was applied to predetermine sample size. All experiments were conducted three times unless stated otherwise. Data that could be reproduced in biological replicates were reported. Statistical analysis for the comparison between two groups was conducted using Student's t-test, while comparisons between multiple groups was conducted using two-way ANOVA post-hoc tests.

**Reporting summary.** Further information on research design is available in the Nature Research Reporting Summary linked to this article.

## Data availability

The ChIP-seq and AmpliSeq datasets generated during the current study have been deposited in GEO with the Accession number GSE129027 and could be accessed at this link: https://www.ncbi.nlm.nih.gov/geo/query/acc.cgi?acc=GSE129027. All the other data supporting the findings of this study are available within the article and its supplementary information files and from the corresponding author upon reasonable request. A reporting summary for this article is available as a Supplementary Information file.

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

## Acknowledgements

We thank Pieter Eichhorn for providing the expression vectors for this study. This work was funded to Boon-Cher Goh by the Singapore Ministry of Health's National Medical Research Council, the National Research Foundation Singapore and the Singapore Ministry of Education under its Research Centres of Excellence initiatives. S.K. and C.S.V. thank A*STAR (including an Industry Alignment Pre-Positioning Grant for the Peptide Engineering Program; grant IDs H17/01/a0/010, IAF111213C) and National Supercomputing Centre Singapore for their support.

## Author contributions

Conceptualization, L.R.K., C.F.C., and B.C.G.; Methodology, L.R.K., R.W.O., G.P., M.L., S.K., C.S.V., D.P.L., H.T.H., and C.F.C.; Software, T.Z.T., J.V.C., and L.Y.J.K.; Validation, L.R.K.; Formal analysis, L.R.K., T.Z.T., J.V.C., and L.Y.J.K.; Investigation, L.R.K., R.W.O., T.Z.T., N.A.B.M.S., M.T., J.A.L., and T.B.U.L.; Resources, G.P., W.J.C., D.P.L., H.T.H., C.F.C., and B.C.G.; Data Curation, L.R.K., T.Z.T., S.K., C.S.V., and L.Y.J.K.; Writing – Original Draft, L.R.K. and B.C.G.; Writing—Review ānd Editing, L.R.K., L.W., M.L., W.J.C., C.S.V., A.V., C.F.C., and B.C.G.; Visualization, L.R.K., M.L., A.V., and C.F.C.; Supervision, B.C.G.; Project Administration, L.R.K. and B.C.G.; Funding acquisition, B.C.G.

## Competing interests

S.K. and C.S.V are the founders of Sinopsee Therapeutics, a biotech company developing molecules for therapeutic purposes. The remaining authors declare no competing interests.
