## [Peer Review File · Nature Communications]

Reviewers' comments:

Reviewer #1 (Remarks to the Author) (expert in lung cancer):

In the manuscript " Targeting codon 158 p53-mutant cancers via the induction of p53 acetylation" Kong et al examined a p53 mutant -R158G common in lung squamous cell carcinoma (LUSC) and used a therapeutic library to determine if they can successfully target the mutant to attack the cancer cells. The authors also suggested a unique mechanism of action through NF- κ B pathway drugs induce cell death via p53-R158G mutant. Using a very limited number of cell lines and only one p53 mutant they attempt to show that NF- κ B pathway intervention happens as a result TRAIIP upregulation by mutant p53. Authors claim that their work would open a new therapeutic line of approach.

Although the work reported has potential, there are two major issues that need to be addressed before one can truly realize how unique the situation is; this will determine whether a new avenue will open. Among many other things to be addressed the authors must (1) examine a number of p53 mutant alleles in different positions in the DNA binding domain as well as different p53 mutations in the same position of 158 without altering cellular context; (2) must examine these mutants under different cellular context in order to generalize any comments they are making (and they are indeed making a number of them).

Following are other significant points that need to be addressed.

1. p3 lines 17-18. "In contrast to the other well-established hotspot mutp53, the functional aspects of this mutation has not been well characterized". This needs a reference validating that lot of work has been done on "the other well-established hotspot mutp53".
2. p3 lines 29-31. "Moreover, 30 p53R158G/H/L was shown to be the most common alteration in LUSC among cases with TP53 mutations (5.5%) (Table S2), as compared to the other hotspot mutants reported among other cancers". This statement is not true when you look at IARC p53 database. 39/30, 273/158. p53-R273H remains the most mutated one. Also, according to AACR Project GENIE registry, R248 has equal frequency of alteration in lung squamous carcinoma as R158.
3. Figure 1A. It is in general difficult to determine how many times an experiment has been repeated. The authors did not mention how long the cells were treated with the drugs?
4. p4 lines 32-33. "In contrast, profound synergistic response was observed in p53R158G cells (Bliss value < 0) when combining belinostat (0.1 μ M) with cisplatin (Figure 1C)." Figure 1D did not show profound synergy between the two drugs in 158 cells especially in clone#8 which was not statistically significant either. There has to be a quantitative way to demonstrate synergy. Visual impression of the westerns does not clearly attest to what the authors claim. MBA-MB-468 (R273H, SK-BR-3 (R175H) particularly showing synergy. They need to use some kind of quantitative program to make their point. Until this observation is not clarified, the entire premise of the work becomes weak. Several cell lines, including PDX or primary cell lines with 158 mutation need to be used to make the point. Until then the statement (p5, lines 4-5) "These findings indicate specific mechanistic underpinnings of the observed synergy in p53R158G cells." Becomes irrelevant.
5. p5 lines 9-11. "...acetylation of histones affecting DNA configuration 10 favoring mutp53 access to DNA binding elements, or direct modification of mutp53 itself leading to alternative DNA binding." There is apparently no direct evidence of direct DNA binding of mutant p53; if there are any references authors need to cite them.
6. p5, line 12. "(Figure 2D)" should read (Figure 2B)".
7. p5, lines 28-30. "Cicplatin and belinostat triggered comparable S-phase arrest regardless of p53 status...(Figure 2H,3B)" Belinostat looks to have no effect on S-phase arrest in either figure, the effect seen looks to be solely because of cisplatin treatment and therefore no synergy.
8. Figure 2D. They show the data with H2170 cell line alone. More cell lines and samples are needed.
9. Figure 2C, PARP no significant difference is visible unfortunately; caspase has some difference, difficult to know if synergy exists.

10. Figure 3A, C. It is surprising that they have not included PXD alone. PXD alone indeed affected acetylation of H3 and H4 histone in H2170 cells in Figure 2. In Figure 3A why is the total p53 level almost non-existent in lane 7? Also, for Figure 3A, why is there no total p53 in the p53WT expressing cells after treatment with cisplatin or in combination with belinostat when p53 can be clearly seen in the same cells in Figure 5F?
11. Figure 3 D-H. Fig 3E for 158 almost no improvement when both are present for CDKN1A. For PUMA it went down and for WT very slight increase. For G, PMAIP1 no increase for 158 and WT. For BAX, some increase for 158 and WT. Data seem to be all over the place. What is a rational explanation?
12. Authors should give scale bars for the images in Figure 3I
13. p6 lines 4-7. "While the modifications were unable to overcome cisplatin damage leading to apoptosis, belinostat-induced synergy was abrogated in p53R158G (DBD, K-A) and p53R158G(K20A) cells (Figure 3A, C), both cell types with impaired Lysine acetylation at the p53 DBD." Does it mean synergy is not needed for apoptosis?
14. Some Figure legends are very poorly explained, E.g., Figure S6A.
15. P8. Line 10. "Two chromatin regulatory genes downstream of ETS2, KMT2D and KAT6A 21, reported to be elevated in hotspot GOF mutp53 (R175H, R248Q, R248W, R249S or R273H), were neither transactivated nor upregulated in p53R158G (Figure 4F,G, Figure S6G,H), thereby distinguishing DNA binding by Arg158 mutants from other mutated p53 GOF isoforms." Is this under DNA damage or under normal condition?
16. In Figure 4H if TRAIIP is transactivated by acetylated mutant p53 why is the protein levels same as p53-/-?
17. Figure S9. p8 lines 24-31. Instead of using mutant p53 overexpressing cell lines they should generate cell lines by CRISPR/Cas9 systems. More p53 mutants need to be tested.
18. p9 lines 9-10 "To rule out the possible alternative involvement of non-canonical NFκB signaling, we investigated the transactivation of p52 (NFκB2 subunit) and showed that DNA-binding was minimally detected in all cell types (data not shown)". Authors need to show transactivation and DNA-binding data.
19. Figure 5G. Expression of exogenous TRAIIP was shown for p53-/- and p53 WT in Figure 5F but not for p53 R158G. TRAIIP expression in that cell type should be shown.
20. p9 lines 27-33. "As previously described 49, ablation of TRAF2 reduced phosphorylation and degradation of IκB (Figure 5F), therefore facilitating IκB-mediated impediment of NFκB transactivation (Figure 5H), and increased drug-induced apoptosis (Figure 5G). Importantly, exogenous TRAIIP recapitulated p53R158G response to drug treatment in p53wt cells. Similar observations were made in H2170, whereby acetylation of p53R158G is associated with p21 upregulation, p-IκB suppression and elevated TRAIIP transcript (Figure S11A,B)." Data is not as clear, since p21 is high when TRAIIP is hardly overexpressed and IκB phosphorylation is pretty high.
21. p9 lines 33-34. "Of note, these changes were not found in other GOF mutp53 cells (MDA-MD-468 and SKBR-3 cells), thereby underlying the distinctiveness of the observed mechanism to the p53R158G mutant." Data is not clear with 468 and skBR3. 468 has p21 over expression. SKBR3 has reduction of TRAIIP. Explanation?
22. p10 lines 29-30. "Mechanistically, acetylation of mutp53R158G was accompanied by elevated TRAIIP (Figure 30 S14E), p65 inhibition (Figure S14F) and reduced cell proliferation (Figure S14G);" A complete growth assay needs to be shown, KI67 labeling is not enough.
23. p10, line 35-p11, line 2. "Effective silencing of TRAIIP markedly rescued cellular apoptosis, as demonstrated by the reduced PARP and caspase 3 cleavage, in H2170 cells by sustaining p-IκB without affecting the p53 acetylation (Figure S11C)." TRAIIP expression doesn't seem to be reduced in DMSO treated cells after siRNA treatment, is there a reason for this?
24. p10, line 25. "When applied in vivo, cisplatin was effective in blocking tumor growth...(Figure S14C)" Since the tumor treated with cisplatin was still growing, changing the word from "blocking" to "reducing" makes more sense.
25. p11 lines 14-16. "Concordant with our in vitro findings, cisplatin and topotecan significantly induced p53 acetylation and phosphorylation, which correlated to TRAIIP upregulation as well as PARP cleavage in tumors (Figure 6G)." PARP cleavage is not always proportional to acetylation and

phosphorylation. Should address.

26. p11 lines 18-22. "These findings suggest that, while the Arg158 p53 mutation is more prevalent in lung carcinomas, the mechanism conferring tumor cytotoxicity through mutp53 acetylation is potentially a general phenomenon in various cancer types harboring the genetic alteration, and therefore amenable to therapeutic intervention through a similar mechanism." The data behind the statement remain very weak. A lot more experiments need to be done before a generalized statement like this can be made.

27. p12 lines 12-18. "The complex crosstalk between p53 and NFκB have been described previously, with the suppression of p65 in wild-type cells a result of competitive interaction with p300 or glucocorticoid receptor; conversely mutp53 prolongs and enhances NFκB activation. We propose here a novel mechanism for this divergent effect through acetylation of the DBD of mutp53R158G, which alters its DNA binding spectrum and upregulates TRAIP as key target gene, leading to NFκB suppression through TRAF2 degradation and culminating in cell death (Figure 7)." There are more authentic and appropriate reference that need to be incorporated on mutant p53-mediated transactivation and mutant p53-Nf-kb relationship.

Upregulation in H2170 at protein level of TRAIP is marginal although at quantitation (QPCR?) shows significant. We need to see such quantitation for other two cell lines too.

They need to do gel retardation to show DNA binding.

28. p13 lines 18-20. "This is unsurprising, as it has been recently recognized that different p53 mutations are attributable to distinctive phenotypes and cellular effects, and our work adds to this." Needs a reference.

29. P14 line 13. They have generated WT p53 overexpressing cell line. Since Overexpression of WT p53 usually leads to cell death or growth blockage, the authors need to check sequence of expressing p53.

30. ChIP seq. How many times each sample have been repeated needs to be clearly stated.

Reviewer #2 (Remarks to the Author) (expert in p53):

In this manuscript by Kong et al, the author proposed a new strategy to treat cancers containing Arg158-mutp53 by combination use of acetylation enhancement of mutp53 and DNA damage drugs. The concept is novel and interesting. The author also found a new target of Arg158-mutp53, which may explain the gain-of-function of this mutant p53. However, the authors need to address the following concerns by experiments before it can acceptable.

1. What are the gain-of-functions for Arg158-mutp53 in addition to acceleration of tumor growth as shown in Figure S1. The authors need to look at more aspects like metastasis. Moreover, the authors should put this data in main figures but not in supplementary figures.

2. The authors claimed that induction of acetylation of mutp53 can alter its function, but they did not prove what specific sites of acetylation are critical in this process. The authors should clarify which acetylation site is decisive in function transformation of Arg158-mutp53.

3. If triggering acetylation also re-activates other mutant p53s such as 175H, 248W, and 273H? The author should compare and give possible explanations.

4. In Figure 2A, the authors claimed that "This synergy was validated in H2170 cells, with belinostat augmenting cisplatin-induced caspase 3 and PARP cleavage", but no significant difference is seen in PARP cleavage.

Generally, CDDP plus PXD101-induced apoptosis seems to be only slightly stronger than treatment

with CDDP alone (e.g., Fig. 2G and Fig. 3C). It would be hard to tell if there is a difference in vivo. Thus, the authors should compare the therapeutic effects between those two groups in animal model.

5. In Figure 3K, why the basal acetyl-p53 level is so high even without Tenovin-6 treatment in p53 R158G cells. In contrast, the basal acetyl-p53 level is null in Figure 3A.

6. Figure 4D, CDDP plus PXD101-induced TRAIP mRNA expression is much higher in p53 R158G cells than in p53 WT and p53 null cells, but Western data (Figure 4H) showed that there was no difference between p53 R158G and p53 null cells. The authors need to repeat the experiment or explain the possible reasons.

Reviewers' comments:

Reviewer #1 (Remarks to the Author) (expert in lung cancer):

In the manuscript "Targeting codon 158 p53-mutant cancers via the induction of p53 acetylation" Kong et al examined a p53 mutant -R158G common in lung squamous cell carcinoma (LUSC) and used a therapeutic library to determine if they can successfully target the mutant to attack the cancer cells. The authors also suggested a unique mechanism of action through NF-kB pathway drugs induce cell death via p53-R158G mutant. Using a very limited number of cell lines and only one p53 mutant they attempt to show that NF-kB pathway intervention happens as a result TRAIIP upregulation by mutant p53. Authors claim that their work would open a new therapeutic line of approach.

Although the work reported has potential, there are two major issues that need to be addressed before one can truly realize how unique the situation is; this will determine whether a new avenue will open. Among many other things to be addressed the authors must (1) examine a number of p53 mutant alleles in different positions in the DNA binding domain as well as different p53 mutations in the same position of 158 without altering cellular context; (2) must examine these mutants under different cellular context in order to generalize any comments they are making (and they are indeed making a number of them). Following are other significant points that need to be addressed.

We thank the Reviewer for this suggestion, which we found insightful to strengthen the findings discussed in the manuscript. Further to the work submitted, we have currently examined:

1) codon 158 p53 mutants in lung cancer cells [**H441 (lung adenocarcinoma, p53R158L)**, **H661 (lung large cell carcinoma, p53R158L)**], and under different cellular context [**H747 (colorectal carcinoma, p53R158L)**].

2) p53 mutant alleles in different positions in the DNA binding domain in lung cancer cells [**lung adenocarcinoma H1417 (p53R175L) and H1975 (p53R273H)**] as well as different p53 mutations in different cellular context [**breast carcinoma SK-BR-3 (p53R175H)**, **HCC70 (p53R248Q)**, **BT-549 (p53R249S)** and **MDA-MB-468 (p53R273H)**; **pancreatic carcinoma MIA-Paca-2 (p53R248W) and PANC-1 (p53R273H)**].

In summary, cell lines with codon 158 mutp53 exhibit more robust apoptotic response (cleaved PARP and Caspase 3) when treated with belinostat/cisplatin combination, as opposed to cisplatin alone. While other GOF mutant cells demonstrate differential sensitivity to cisplatin-induced apoptosis, this effect is not potentiated by the addition of belinostat.

The findings have been added as Figure 3A and Supplementary Figure 3 in the revised manuscript (Pg 5, line 7-20).

1. p3 lines 17-18. "In contrast to the other well-established hotspot mutp53, the functional aspects of this mutation has not been well characterized". This needs a reference validating that lot of work has been done on "the other well-established hotspot mutp53".

References on other hotspot *TP53* mutants have been added to the manuscript (Lang GA, et al. Cell 119, 861-872 (2004), Hanel W, et al. Cell Death Differ 20, 898-909 (2013), Olive KP, et al. Cell 119, 847-860 (2004), Di Agostino S, et al. Cancer Cell 10, 191-202 (2006), Liu D, et al. Oncogene 29(7):949-956 (2010), Zhu J, et al. Nature 525, 206-211 (2015)).

2. p3 lines 29-31. "Moreover, 30 p53R158G/H/L was shown to be the most common alteration in LUSC among cases with TP53 mutations (5.5%) (Table S2), as compared to the

other hotspot mutants reported among other cancers”. This statement is not true when you look at IARC p53 database. 39/30, 273/158. p53-R273H remains the most mutated one. Also, according to AACR Project GENIE registry, R248 has equal frequency of alteration in lung squamous carcinoma as R158.

We thank the reviewer for the effort to search the IARC p53 database and the AACR databases and for pointing this out. The initial analysis was conducted with TCGA database which led to this statement. In the light of data from other databases, we have amended the statement to: “Among the *TP53* mutations found in ~50% of non-small cell lung cancer, p53^{R158G/H/L} is one of the most common *p53* mutation hotspots according to multiple public databases (TCGA, COSMICS, IARC p53 Database) (Pg 3, line 27-29), despite being reported in different frequencies.”

3. Figure 1A. It is in general difficult to determine how many times an experiment has been repeated. The authors did not mention how long the cells were treated with the drugs?

We apologise for the lack of clarity in the description of this figure. The drug screening was performed once with technical triplicates. Selected targets were then validated independently (n = 3). Most treatments, unless otherwise specified, were performed for 48h (for immunoblotting, imaging and NFkB activity) or 72h (For cell viability). The details have been provided in the figure captions and Methods.

4. p4 lines 32-33. “In contrast, profound synergistic response was observed in p53R158G cells (Bliss value < 0) when combining belinostat (0.1 μM) with cisplatin (Figure 1C).” Figure 1D did not show profound synergy between the two drugs in 158 cells especially in clone#8 which was not statistically significant either. There has to be a quantitative way to demonstrate synergy. Visual impression of the westerns does not clearly attest to what the authors claim. MBA-MB-468 (R273H, SK-BR-3 (R175H) particularly showing synergy. They need to use some kind of quantitative program to make their point.

We thank the Reviewer for the comment. We will like to point out that the synergistic measurement reported in Figure 2C has been conducted and quantified based on Bliss combination model, which is a widely used platform to screen for candidate drug combinations, **by comparing the observed combination response with the predicted combination response**. By assuming that there is not drug-drug interaction, the combination effect is declared synergistic if the observed response is greater than the predicted response (in this case, the sum of growth inhibition by belinostat alone and cisplatin alone). Moreover, data presented in Figure 2D compared cisplatin IC50 in various p53 isogenic clones, in which a sublethal dose of belinostat significantly left-shifted the dose-response curve of 4/5 p53R158G clones. The lack of synergy in clone 8 may likely be due to clonal selection.

In order to provide better clarity of this analyses, we have:

1) provided a detailed indication and explanation of Bliss Independence model in the manuscript (Methods, Pg 17, line 22), and

2) performed densitometric quantification of cleaved PARP and caspase 3 blots for various mutp53 cell lines (Figure 3B, Supplementary Figure 3C-D) and various p53 isogenic clones (Figure 4B).

Until this observation is not clarified, the entire premise of the work becomes weak. Several cell lines, including PDX or primary cell lines with 158 mutation need to be used to make the

point. Until then the statement (p5, lines 4-5) “These findings indicate specific mechanistic underpinnings of the observed synergy in p53R158G cells.” Becomes irrelevant.

We totally agree with the Reviewer on this. We hope that with the further analyses, we have strengthened our claim on the synergism induced by belinostat and cisplatin. Efforts have been made to obtain primary cell lines harbouring p53 codon 158 mutations, but so far our team was unable to obtain sufficient biopsy samples to do that. In order to address this and validate the synergy, we have 1) established several isogenic p53R158G clones, 2) established shp53 knockdown stable clones from codon 158 mutant H2170 cells. In addition, we have repeated the experiments in several codon 158 mutp53 tumours in various cellular context (Figure 3A).

Taken together, the newly acquired data suggest that apoptosis associated with p53 acetylator is profoundly observed in codon 158 mutant cells of various cellular contexts. This is in concordant with our already shown *in vivo* data, whereby PDX models of gastric and hepatocellular tumors harbouring Arg158 mutp53 were effectively targeted by topotecan and cisplatin; and our data demonstrated that both compounds are effective acetylators of mutp53 *in vivo*.

5. p5 lines 9-11. “...acetylation of histones affecting DNA configuration 10 favoring mutp53 access to DNA binding elements, or direct modification of mutp53 itself leading to alternative DNA binding.” There is apparently no direct evidence of direct DNA binding of mutant p53; if there are any references authors need to cite them.

We thank the Reviewer for commenting on the role of mutp53. There is currently emerging evidence in the field regarding the DNA binding capacity of mutp53, just to list a few here (Friedlander P, et al. *Journal of Biological Chemistry*. 271(41):25468–25478 (1996); Gohler T, et al. *Nucleic Acids Res*. 33(3): 1087–1100 (2015); Brázdová M, et al. *PLOS ONE* 8(3): e59567 (2013). The functional aspect of hotspot mutp53 DNA binding has been thoroughly investigated in Zhu J, et al. *Nature* 525, 206-211 (2015), which described how DNA binding of mutp53 mutants co-opt chromatin pathways to drive cancer growth. This paper has been cited in our manuscript.

These findings from other researchers, coupled with the report that acetylation at C-terminal is crucial for DNA binding of p53 (Gu W, et al. *Cell* 90, 595-606 (1997)), have led us to draw the hypothesis stated above.

6. p5, line 12. “(Figure 2D)” should read (Figure 2B)”.

This has been amended in the revised manuscript.

7. p5, lines 28-30. “Cisplatin and belinostat triggered comparable S-phase arrest regardless of p53 status...(Figure 2H,3B)” Belinostat looks to have no effect on S-phase arrest in either figure, the effect seen looks to be solely because of cisplatin treatment and therefore no synergy.

We apologise for the confusion in this statement. The statement was supposed to be “**Cisplatin alone and in combination with belinostat triggered comparable S-phase arrest regardless of p53 status...**”. This is to support our claim that “the synergy is independent of its cytostatic effect”, and indeed apoptosis is induced as shown. We have amended this in the revised manuscript (Pg 6, line 9-10).

8. Figure 2D. They show the data with H2170 cell line alone. More cell lines and samples are needed.

We have added in the investigation of p53 PTM in several cell lines with different p53 mutational status. The data have been added (Figure 3A, Supplementary Figure 3A,B and 12A) and described accordingly in the text (Pg 5, line 10-13). All these codon 158 p53 mutant cell lines in different malignancies show synergistic sensitivity to the cisplatin/belinostat combination compared to cisplatin alone.

9. Figure 2C, PARP no significant difference is visible unfortunately; caspase has some difference, difficult to know if synergy exists.

We thank the Reviewer for pointing this out. We have performed densitometric quantification of the PARP and caspase 3 blots (Supplementary Figure 3C-D). The analyses have revealed that combination with belinostat had triggered the cleavage of these apoptotic biomarker at lower cisplatin doses in H2170 cells. At high combination doses, the elevated cell death was demonstrated by the significant drop in un-cleaved PARP as compared to cisplatin treatment alone. The lack of clear obvious increase in PARP cleavage is likely due to saturation of protein expression.

10. Figure 3A, C. It is surprising that they have not included PXD alone. PXD alone indeed affected acetylation of H3 and H4 histone in H2170 cells in Figure 2. In Figure 3A why is the total p53 level almost non-existent in lane 7? Also, for Figure 3A, why is there no total p53 in the p53WT expressing cells after treatment with cisplatin or in combination with belinostat when p53 can be clearly seen in the same cells in Figure 5F?

We agree with the Reviewer that belinostat alone significantly induce H3/H4 acetylation. However, the objective for this immunoblotting experiment was to compare the extent of p53 acetylation of cisplatin-treated cells, in the absence/presence of low dose belinostat that had been shown to have minimal effect on either p53 PTM or apoptosis (Figure 3C,D, Figure 6G).

In Figure 4A (previously Figure 3A), there was indeed expression of p53 in the mutp53 control (lane 7) and wtp53 treated (lane 5-6) cells, but a blot with low exposure was presented to present the optimal exposure for mutp53. We have included the corresponding blot with higher exposure that clearly shown increase of p53 in the wtp53 cells.

11. Figure 3 D-H. Fig 3E for 158 almost no improvement when both are present for CDKN1A. For PUMA it went down and for WT very slight increase. For G, PMAIP1 no increase for 158 and WT. For BAX, some increase for 158 and WT. Data seem to be all over the place. What is a rational explanation?

We thank the Reviewer for this comment.

We will like to clarify the following:

1) cisplatin-induced p53 acetylation is more prominent in mutp53 cells, and could be further induced by combination of cisplatin/belinostat (Figure 4I),

2) the qPCR data showed that mutp53 partially retain the capability to transactivate p53 downstream genes (CDKN1A, PUMA, BAX),

3) we propose that cell death associated with mutp53 is associated with the negative regulation of TRAF2/NFkB signalling through TRAIIP transactivation.

Taken together, we reason that p53R158G possesses DNA-binding ability (which we further demonstrated using reporter assay). This is validated in our later analyses of the ChIP-seq and transcriptome (Figure 5B) that *PMAIP1* and *CDKN1A* could be transactivated by mutp53, but to a lesser extent than wtp53. Our structural simulation predicted that “acetylation of the Lysine residues within the DBD reduces the sidechain positive charges, resulting in structural and chemical interactive alterations which restore the dimerization capacity of the core domain of mutp53R158G” (Pg 7, line 15). This then partially recover wild type p53 function. In addition, we show that co-treatment with belinostat significantly elevates transcription of TRAIIP (Figure 5D, Supplementary Figure 10). Therefore, we postulate that acetylation of mutp53 could lead to 1) restoration of wtp53 binding (thus partial increase in p53 downstream genes) and 2) significant increase in TRAIIP suppress NFκB-associated survival signal, thus potentiating cisplatin-induced apoptosis.

12. Authors should give scale bars for the images in Figure 3I

We apologise for the oversight. The scales bars have been included in the revised manuscript.

13. p6 lines 4-7. “While the modifications were unable to overcome cisplatin damage leading to apoptosis, belinostat-induced synergy was abrogated in p53R158G (DBD, K-A) and p53R158G(K20A) cells (Figure 3A, C), both cell types with impaired Lysine acetylation at the p53 DBD.” Does it mean synergy is not needed for apoptosis?

Cisplatin forms adducts that crosslink DNA strands leading to cell death through DNA damage response, and our data (cell cycle arrest, DNA damage markers, comet assay) suggested that this is not affected by belinostat co-treatment. As cisplatin is known to induce apoptosis by activating the extrinsic apoptotic pathway that are p53-independent, this may explain the single agent activity of cisplatin. Nonetheless, with the combination of belinostat/cisplatin in mutp53 cells, it is clear that an alternative mechanism involving acetylation of mutp53 is active; therefore, p53^{R158G (DBD, K-A)} and p53^{R158G(K20A)} cells which cannot be acetylated at these respective sites will abrogate this pathway of apoptosis

14. Some Figure legends are very poorly explained, E.g., Figure S6A.

We have made changes in the Figure legends to provide better descriptions of the figures, particularly for Figure 5A and Supplementary Figure S7A.

15. P8. Line 10. “Two chromatin regulatory genes downstream of ETS2, KMT2D and KAT6A 21, reported to be elevated in hotspot GOF mutp53 (R175H, R248Q, R248W, R249S or R273H), were neither transactivated nor upregulated in p53R158G (Figure 4F,G, Figure S6G,H), thereby distinguishing DNA binding by Arg158 mutants from other mutated p53 GOF isoforms.” Is this under DNA damage or under normal condition?

These are the basal mRNA expression of KMT2D and KAT6A under untreated conditions. Based on the report by Zhu J, et al. (Nature 2015), both of these genes are positively regulated by GOF p53, leading to higher expression in mutp53 cells compared to wtp53 and p53 knockout cells.

16. In Figure 4H if TRAIIP is transactivated by acetylated mutant p53 why is the protein levels same as p53-/-?

We thank the Reviewer for pointing this out, and we agree with the observations. The authors have discussed internally and believe that **TRAIP upregulation could be a global response to DNA damage, and the loss of p53** as a checkpoint control likely lead to a greater DNA damage signal. It has been reported that TRAIP is associated with genotoxic lesions during genome replication, leading to H2AX phosphorylation upon replication-associated DNA damage (Harley ME, et al. Nat Genet. Jan; 48(1): 36–43. (2016)). TRAIP could be activated in cancer cells in response to cisplatin-induced DNA intercalation independently of p53. Upon DNA damage (with cisplatin), TRAIP is highly expressed in p53R158G cells and leads to NFκB suppression, and based on our immunoblotting data in Figure 5H, p-IκB was only suppressed in mutp53 cells. Accordingly, we postulate that TRAIP could be transactivated in mutp53 cells as a negative regulator to NFκB signalling through its E3 ubiquitin ligase function. This train of thought has been incorporated into the corresponding Result section in the revised manuscript (Pg 9, line 6).

17. Figure S9. p8 lines 24-31. Instead of using mutant p53 overexpressing cell lines they should generate cell lines by CRISPR/Cas9 systems. More p53 mutants need to be tested.

We thank the Reviewer for pointing this out. We are aware of the possible study models to study these mutant p53, and consultation with other authors have led us to conclude that isogenic cell lines generated from a p53 null background (in our case, Calu-1) are suitable to conduct p53-related experiments. We also take into consideration the effectiveness to generate multiple acetylation-defective models.

We have carried out several key experiments with conventional mutp53 cells harbouring different GOF hotspot mutations. The additional data have been included in Supplementary Figure 12.

18. p9 lines 9-10 “To rule out the possible alternative involvement of non-canonical NFκB signaling, we investigated the transactivation of p52 (NFκB2 subunit) and showed that DNA-binding was minimally detected in all cell types (data not shown)”. Authors need to show transactivation and DNA-binding data.

The data for p52 (NFκB2 subunit) transactivation (TransAM assay) has been added into the revised manuscript (Figure 6B).

19. Figure 5G. Expression of exogenous TRAIP was shown for p53^{-/-} and p53 WT in Figure 5F but not for p53 R158G. TRAIP expression in that cell type should be shown.

The Western blots reflecting exogenous TRAIP expression in each cell types in Figure 6I (previously Figure 5G) have been shown (Figure 6J). We have also provided a PARP cleavage blot to complement the Annexin V staining shown in Figure 6I, as supplemented in Pg 10, line 19-21).

20. p9 lines 27-33. “As previously described 49, ablation of TRAF2 reduced phosphorylation and degradation of IκB (Figure 5F), therefore facilitating IκB-mediated impediment of NFκB transactivation (Figure 5H), and increased drug-induced apoptosis (Figure 5G). Importantly, exogenous TRAIP recapitulated p53R158G response to drug treatment in p53wt cells. Similar observations were made in H2170, whereby acetylation of p53R158G is associated with p21 upregulation, p-IκB suppression and elevated TRAIP transcript (Figure S11A,B).” Data is not as clear, since p21 is high when TRAIP is hardly overexpressed and IκB

phosphorylation is pretty high.

We thank the Reviewer for this comment. Our data suggest that cisplatin and combination treatment induced TRAIIP expression and suppression of p-I κ B in p53-expressing cells, and we validated these observations in cell lines harbouring various p53 hotspot mutations (Supplementary Figure 12A). In addition, we are providing densitometry for TRAIIP blots in Supplementary Figure 12A as supporting evidence that TRAIIP is upregulated. We also agree that the level of p-I κ B may vary across cell lines, but this is likely to be cell-context dependent.

21. p9 lines 33-34. “Of note, these changes were not found in other GOF mutp53 cells (MDA-MD-468 and SKBR-3 cells), thereby underlying the distinctiveness of the observed mechanism to the p53R158G mutant.” Data is not clear with 468 and skBR3. 468 has p21 over expression. SKBR3 has reduction of TRAIIP. Explanation?

We thank the Reviewer for this comment. In order to generalize the mechanistic interference of NF κ B across cancer cells, we have repeated the experiments with several other cell lines harbouring p53R158G/H/L or other p53 hotspot mutations (Supplementary Figure 12A). The data collectively show that cisplatin alone or in combination with belinostat were able to induce acetylation in codon 158 mutant cells (H441, H661, H747), and concordantly leading to TRAIIP upregulation (Supplementary Figure 12B) and apoptosis. In contrast, this mechanism is not demonstrated in non-codon 158 mutant cell lines. These data are supplemented in Pg 10, line 22-27.

22. p10 lines 29-30. “Mechanistically, acetylation of mutp53R158G was accompanied by elevated TRAIIP (Figure 30 S14E), p65 inhibition (Figure S14F) and reduced cell proliferation (Figure S14G);” A complete growth assay needs to be shown, KI67 labeling is not enough.

We will like to highlight that the tumour growth was monitored and the data shown in Supplementary Figure 15C. We have also amended the sentence to clarify the statement in the revised manuscript (Pg 11, line 21).

23. p10, line 35-p11, line 2. “Effective silencing of TRAIIP markedly rescued cellular apoptosis, as demonstrated by the reduced PARP and caspase 3 cleavage, in H2170 cells by sustaining p-I κ B without affecting the p53 acetylation (Figure S11C).” TRAIIP expression doesn’t seem to be reduced in DMSO treated cells after siRNA treatment, is there a reason for this?

We agree with the Reviewer on this, and we are fully aware of this unusual observations. The experiment had been repeated (n = 3), and we consistently observed the expression of TRAIIP protein in the DMSO-treated cells upon siRNA transfection. We reason that the presence of a compensatory mechanism could potentially stabilise TRAIIP and counterbalance the knockdown effect, and this mechanism may be impaired under cellular stress (eg drug treatment). However, this has not been explored further and the precise mechanism remains unclear.

24. p10, line 25. “When applied in vivo, cisplatin was effective in blocking tumor growth...(Figure S14C)” Since the tumor treated with cisplatin was still growing, changing

the word from “blocking” to “reducing” makes more sense.

This has been amended in the revised manuscript (Pg 11, line 16).

25. p11 lines 14-16. “Concordant with our in vitro findings, cisplatin and topotecan significantly induced p53 acetylation and phosphorylation, which correlated to TRAIIP upregulation as well as PARP cleavage in tumors (Figure 6G).” PARP cleavage is not always proportional to acetylation and phosphorylation. Should address.

We agree with the Reviewer to a certain extent that PARP cleavage is not always proportional to p53 acetylation. This is likely due to the other effects of the compounds, for instance, topotecan is a chemotherapeutic agent targeting topoisomerase which may generate genotoxicity that induce p53 modification. In addition, the heterogeneity of PDX tumours may also contribute to this variation. Overall, our observations indicate a positive correlation of acetylated p53 with PARP cleavage. We have amended the sentence (Pg 12, line 7-8) to clarify this.

26. p11 lines 18-22. “These findings suggest that, while the Arg158 p53 mutation is more prevalent in lung carcinomas, the mechanism conferring tumor cytotoxicity through mutp53 acetylation is potentially a general phenomenon in various cancer types harboring the genetic alteration, and therefore amenable to therapeutic intervention through a similar mechanism.” The data behind the statement remain very weak. A lot more experiments need to be done before a generalized statement like this can be made.

We thank the Reviewer for the suggestion. We have expanded our investigations by conducting similar experiments on:

- 1) other hotspot mutp53 cells (codon 175, 248, 273),
- 2) lung carcinoma cells with p.R158L mutation (H441 and H661), and
- 3) cancer cells from other organ harbouring codon 158 mutation (H747, caecum) mutations.

The additional data (Figure 3A and Supplementary Figure 3) suggest that:

- 1) mutp53-TRAIIP-NFκB regulation confers cytotoxicity in codon 158 mutant cells,
- 2) In addition to lung cancers bearing the p53R158G mutation, the same mechanistic activation of TRAIIP by acetylation of p53 is applicable to various codon 158 mutations in a variety of other cancer types.

27. p12 lines 12-18. “The complex crosstalk between p53 and NFκB have been described previously, with the suppression of p65 in wild-type cells a result of competitive interaction with p300 or glucocorticoid receptor; conversely mutp53 prolongs and enhances NFκB activation. We propose here a novel mechanism for this divergent effect through acetylation of the DBD of mutp53R158G, which alters its DNA binding spectrum and upregulates TRAIIP as key target gene, leading to NFκB suppression through TRAF2 degradation and culminating in cell death (Figure 7).” There are more authentic and appropriate reference that need to be incorporated on mutant p53-mediated transactivation and mutant p53-Nf-kb relationship.

The references for this statement have been reviewed in the revised manuscript (Di Minin G, et al. Mol Cell 56, 617-629 (2014), Gulati AP, et al. Mol Carcinog 45, 26-37 (2006)).

Upregulation in H2170 at protein level of TRAIP is marginal although at quantitation (QPCR?) shows significant. We need to see such quantitation for other two cell lines too.

TRAIP blots at Supplementary Figure 12A has been quantified and presented as Supplementary Figure 12B in the revised manuscript.

They need to do gel retardation to show DNA binding.

We thank the Reviewer for the suggestion. Based on our understanding, cellular reporter assay and gel retardation assay (EMSA) are common techniques used to study protein-nucleic acid interaction, but EMSA assay is prone to false positive results. To address the DNA binding ability of mutp53, we have conducted reporter assays on mutp53R158G cells to indicate the binding of mutp53 to the promoter of *CDKN1A* (Supplementary Figure 4A).

28. p13 lines 18-20. "This is unsurprising, as it has been recently recognized that different p53 mutations are attributable to distinctive phenotypes and cellular effects, and our work adds to this." Needs a reference.

References on the unique features of individual hotspot *TP53* mutants have been added to the manuscript (Freed-Pastor WA, Prives C. *Genes Dev* 26, 1268-1286 (2012), Turrell FK, et al. *Genes Dev* 31, 1339-1353 (2017)).

29. P14 line 13. They have generated WT p53 overexpressing cell line. Since Overexpression of WT p53 usually leads to cell death or growth blockage, the authors need to check sequence of expressing p53.

This is a good suggestion. We have sequenced the ORF of the TP53 in the isogenic cell lines to ensure that there is no random mutation in the wtp53 clone. We will like to mention that multiple selection processes have been conducted to isolate this one stable wtp53 clone. Moreover, subsequent qPCR analyses on the wtp53 downstream targets have validated that the TP53 overexpressed in this clone is functionally intact (strong induction of MDM2, etc). The sequencing data have been included as Supplementary Figure 1C-H, and described accordingly (Pg 4, line 2-5).

30. ChIP seq. How many times each sample have been repeated needs to be clearly stated.

The pair-end ChIP-Seq was performed once, and validated independently with ChIP-qPCR (n = 3). The Methods have been modified accordingly to include these information (Pg 16, line 18).

Reviewer #2 (Remarks to the Author) (expert in p53):

In this manuscript by Kong et al, the author proposed a new strategy to treat cancers containing Arg158-mutp53 by combination use of acetylation enhancement of mutp53 and DNA damage drugs. The concept is novel and interesting. The author also found a new target of Arg158-mutp53, which may explain the gain-of-function of this mutant p53. However, the

authors need to address the following concerns by experiments before it can be acceptable.

1. What are the gain-of-functions for Arg158-mutp53 in addition to acceleration of tumor growth as shown in Figure S1. The authors need to look at more aspects like metastasis. Moreover, the authors should put this data in main figures but not in supplementary figures.

We thank the Reviewer for the suggestion. We have performed additional experiments, namely invasion assay and colony forming assay, to support the GOF of codon 158 mutp53:

- 1) comparison of GOF in isogenic Calu-1 cells with different p53 status, and
- 2) comparison of GOF in H2170 cells after TP53 knockdown.

These data have been compiled with our initial Supplementary Figure 1 and included as part of the main figures (Figure 1) in the revised manuscript (Pg 4, line 8-12). In summary, p53^{R158G}-expressing cells are indeed more oncogenic, as demonstrated by the increase in cellular invasion and malignant colony forming ability. In contrast, expression of wtp53 markedly reduced invasiveness and suppressed colony formation. The stark differences in these isogenic cell lines convincingly describe the GOF of codon 158 mutp53.

2. The authors claimed that induction of acetylation of mutp53 can alter its function, but they did not prove what specific sites of acetylation are critical in this process. The authors should clarify which acetylation site is decisive in function transformation of Arg158-mutp53.

In order to address this, we have performed site-directed mutagenesis to generate acetylation-defective mutant of mutp53. We deduced that having each lysine residues to be substituted individually – while technically possible – will complicate data analysis, and thus individual domains were modified instead. Our data suggested that acetylation on the DNA-binding domain is crucial for the observed synergy. Furthermore, molecular modelling was performed to compare the structural dynamics of wild-type p53, mutp53R158G and acetylated-mutp53R158G using monomers of the respective DBD (Supplementary Figure 6). Our simulation predicts that “the formation of a key stabilizing interaction between the polar groups introduced by the acetylated **K101 and K164** that stabilizes the overall conformation of the mutp53” (Pg 7, line 22-27). We postulate that these lysine residues on DBD, codon 101 and 164, are integral to the functional transformation of Arg158-mutp53.

3. If triggering acetylation also re-activates other mutant p53s such as 175H, 248W, and 273H? The author should compare and give possible explanations.

We thank the Reviewer for this comment. Similar experiments have been conducted in our initial draft on two breast cancer cells with hotspot mutp53: SK-BR-3 (p.R175H) and MDA-MD-468 (p.R273H). These data could be located at Supplementary Figure 3A. In the revised manuscript, we have further investigated the effect of belinostat/cisplatin treatment on an extended list of GOF mutp53 cell lines (Supplementary Figure 12A).

Overall, acetylation of mutp53 could be observed in most cell lines (with the exception of H1417 and H1975); but chemosensitizing efficacy of belinostat could only be observed in codon 158 mutant cells in which p53 acetylation is correlated to cisplatin or belinostat/combo treatment. In addition, our structural prediction suggested that Arg158 mutp53 undergoes unique conformational changes when acetylated, with the interactions between acetylated K201 and the minor groove compensating the impaired interaction between K120 and DNA major groove in the mutant isoform (Figure S6D). These observations collectively lead us to draw the conclusion that “This is unsurprising, as it has

been recently recognized that different p53 mutations are attributable to distinctive phenotypes and cellular effects” (Pg 14, line 21-23). This is also concordant with the gaining knowledge in the field that each p53 mutant isoform carries unique GOF mechanism.

4. In Figure 2A, the authors claimed that “This synergy was validated in H2170 cells, with belinostat augmenting cisplatin-induced caspase 3 and PARP cleavage”, but no significant difference is seen in PARP cleavage. Generally, CDDP plus PXD101-induced apoptosis seems to be only slightly stronger than treatment with CDDP alone (e.g., Fig. 2G and Fig. 3C). It would be hard to tell if there is a difference *in vivo*. Thus, the authors should compare the therapeutic effects between those two groups in animal model.

As we have addressed in point 9 of Reviewer 1’s comments, , we have performed densitometric quantification of cleaved PARP and caspase 3 blots (Supplementary Figure 3C-D) to support our claim. We will also like to emphasize that belinostat as a low-dose single agent demonstrated negligible cytotoxicity and cytostatic effect in most cancer cell lines tested. The same dose of belinostat, when used in combination with cisplatin, could elevate cellular apoptosis. By Bliss Independence analyses, the combination of 0.1 μ M belinostat is synergistic with cisplatin (0.1 – 10 μ M).

The usage of belinostat *in vivo* has indeed been explored in several pilot studies, however limited efficacy was observed in these mouse models. As reported in our earlier study (Kong LR, et al. Mol Oncol. Aug;11(8):965-980 (2017)), belinostat exerts poor pharmacokinetic in mouse models with poor pharmacological effect on tumours (lack of histone acetylation). These observations have led us to explore other pharmacologically feasible acetylators of mutant p53 in our *in vivo* experiments.

5. In Figure 3K, why the basal acetyl-p53 level is so high even without Tenovin-6 treatment in p53 R158G cells. In contrast, the basal acetyl-p53 level is null in Figure 3A.

We would like to clarify this experiment. The blot in Figure 3K was set at long exposure to capture the optimal acetylated p53 in both WT and mutant isoforms. Given the intact MDM2-p53 mechanism, the expression of p53 in the WT cells is significantly lower thus making it difficult to detect its acetylated form. In addition, tenovin-6 is a weak acetylator of p53 as compared to cisplatin, thus a longer exposure was required. We have attached a blot with lower exposure in the revised manuscript (Figure 4K).

6. Figure 4D, CDDP plus PXD101-induced TRAIP mRNA expression is much higher in p53 R158G cells than in p53 WT and p53 null cells, but Western data (Figure 4H) showed that there was no difference between p53 R158G and p53 null cells. The authors need to repeat the experiment or explain the possible reasons.

We thank the Reviewer (and Reviewer 1) for pointing this out, and we agree with the observations. It was noted in our qPCR analysis that TRAIP could be induced by belinostat/cisplatin combination in all three p53 isogenic cells, with p53R158G cells demonstrating the most remarkable changes. We will like to point out that TRAIP is known to be involved in cellular response to genotoxic lesions during genome replication, leading to H2AX phosphorylation upon replication-associated DNA damage (Harley ME, et al. Nat Genet. Jan; 48(1): 36–43. (2016)). It is likely that TRAIP is activated in these cell lines as a

result of cisplatin-induced DNA intercalation through in a p53-independent manner, particular in null cells whereby the DNA-damage-repair axis is impaired. Importantly, our findings have associated TRAIIP in mutp53R158G cells to NFkB suppression, and based on our immunoblotting data in Figure 5H, p-IkB was only affected in mutp53 cells. We postulate that TRAIIP could be upregulated by mutp53 as part of a negative regulation to NFkB signalling through its E3 ubiquitin ligase function. This explanation has been incorporated into the corresponding Result section in the revised manuscript (Pg 9, line 6-10).

Reviewers' comments:

Reviewer #1 (Remarks to the Author):

August 9, 2019

- In the first review the major need was to "(1) examine a number of p53 mutant alleles in different positions in the DNA binding domain as well as different p53 mutations in the same position of 158 without altering cellular context; (2) must examine these mutants under different cellular context in order to generalize any comments they are making (and they are indeed making a number of them)." The authors should have used the CRISPR/Cas9 system to generate the mutant lines. Instead, they have chosen an imperfect way of different cell lines with different genetic contexts. This (the imperfection of their cell systems) must be clearly acknowledged as a drawback of the work.

- This reviewer could not find a place in the manuscript where Supplementary Figure 3 has been referred.

Following are other significant points that need to be addressed.

- p10, line 35-p11, line 2. "Effective silencing of TRAIP markedly rescued cellular apoptosis, as demonstrated by the reduced PARP and caspase 3 cleavage, in H2170 cells by sustaining p-IkB without affecting the p53 acetylation (Figure S11C)." TRAIP expression doesn't seem to be reduced in DMSO treated cells after siRNA treatment, is there a reason for this? The authors replied - "We agree with the Reviewer on this, and we are fully aware of this unusual observations. The experiment had been repeated (n = 3), and we consistently observed the expression of TRAIP protein in the DMSO-treated cells upon siRNA transfection. We reason that the presence of a compensatory mechanism could potentially stabilise TRAIP and counterbalance the knockdown effect, and this mechanism may be impaired under cellular stress (eg drug treatment). However, this has not been explored further and the precise mechanism remains unclear. --- This statement should be introduced in the text.

p12 lines 12-18. "The complex crosstalk between p53 and NFκB have been described previously, with the suppression of p65 in wild-type cells a result of competitive interaction with p300 or glucocorticoid receptor; conversely mutp53 prolongs and enhances NFκB activation. We propose here a novel mechanism for this divergent effect through acetylation of the DBD of mutp53R158G, which alters its DNA binding spectrum and upregulates TRAIP as key target gene, leading to NFκB suppression through TRAF2 degradation and culminating in cell death (Figure 7)." They need to do gel retardation to show DNA binding. We thank the Reviewer for the suggestion. Based on our understanding, cellular reporter assay and gel retardation assay (EMSA) are common techniques used to study protein-nucleic acid interaction, but EMSA assay is prone to false positive results. To address the DNA binding ability of mutp53, we have conducted reporter assays on mutp53R158G cells to indicate the binding of mutp53 to the promoter of CDKN1A (Supplementary Figure 4A).--- This is a reporter assay, a functional assay and not a DNA binding assay. They must do the DNA binding (gel retardation, for example and not the reporter assay) assay if they want to show DNA binding.

Reviewer #2 (Remarks to the Author):

The revised manuscript is significantly improved although there are still a few points below that need to be addressed.

1. In rebuttal letter, the authors claimed that "It is likely that TRAIP is activated in these cell lines as a result of cisplatin-induced DNA intercalation through in a p53-independent manner, particular in null cells whereby the DNA-damage-repair axis is impaired."

Unless the authors can see a much stronger induction of TRAIP by PXD+CDDP in p53R158G cells

compared to p53 null cells (Fig. 5H), it is hard to tell if the induction of TRAIP is because of acetylated p53R158G or because of p53-independent factors. The authors should use multiple clones but not just a single clone to prove their hypothesis in Fig. 5H.

2. Combination use of acetylation promoting reagents and DNA damage drugs in treating p53R158G tumor is the key point and major novelty of this manuscript, but the authors did not carry out any in-vivo combination experiments to prove if this strategy works in vivo. The authors claimed that "belinostat exerts poor pharmacokinetic in mouse models with poor pharmacological effect on tumours (lack of histone acetylation)." Further analyses of the combination (single or combined) of these compounds and other DNA damage reagents on p53-mutant tumors may be useful.

Reviewers' comments:

Reviewer #1 (Remarks to the Author):

- In the first review the major need was to “(1) examine a number of p53 mutant alleles in different positions in the DNA binding domain as well as different p53 mutations in the same position of 158 without altering cellular context; (2) must examine these mutants under different cellular context in order to generalize any comments they are making (and they are indeed making a number of them).” The authors should have used the CRISPR/Cas9 system to generate the mutant lines. Instead, they have chosen an imperfect way of different cell lines with different genetic contexts. This (the imperfection of their cell systems) must be clearly acknowledged as a drawback of the work.

We agree with the reviewer that not having genome edited cell lines posed to be a drawback of the study, but we would like to clarify that several attempts have been conducted prior to our initial submission using two different protocols: **1)** Cas9-crRNA-tracrRNA system (Paix, A., Rasoloson, D., Folkmann, A., & Seydoux, G. (2019). *Current Protocols in Molecular Biology*, 129, e102) and **2)** standard CRISPR-Cas9 protocol (Le, C., Ran, FA. et al. (2013) *Science* Feb 15;339(6121):819-23). Unexpectedly, none of the selected clones have been successfully modified with mutant p53. As a result, isogenic clones were generated instead using the p53 null Calu-1 as an alternative approach. Upon receipt of the revised comments, we have tried it a third time with a more robust, marker-free co-selection protocol (Agudelo, D., Durringer, A., Bozoyan, L., et al. (2017) *Nature Methods* 14;615–620) on H2170, A549 and HCT116 cells. Surprisingly, while the system worked well in our parallel knock-out experiments, none of the selected clones was inserted with the HDR template designed to alter *TP53*. We believe this could be due to the nature of the p53 molecule in these cell types. Firstly, it has been reported that wtp53 (in the case of A549 and HCT116) could inhibit CRISPR-Cas9 editing by exerting toxicity when double-strand breaks are detected (Ihry, R.J., Worringer, K.A., et al. (2018) *Nature Medicine* 24;939–946). Secondly, H2170 may have acquired dependency on the GOF mutp53 thus preventing the homozygous allele from being edited.

While we acknowledge the importance of CRISPR-Cas9 system in the context of our study, we believe that the existing models comprising of 1) isogenic cell lines and 2) multiple well-established cell lines commonly used in p53 research, are appropriate to derive our conclusions. In addition, we have discussed the limitation of our models in our discussion, Pg 14, line 25-28 “***One possible caveat of the study is the lack of a universal system to distinguish the functions of the various GOF mutp53 within the same cellular context. The inability to establish a robust CRISPR-Cas9 model has prevented us from studying this.***”

- This reviewer could not find a place in the manuscript where Supplementary Figure 3 has been referred.

The discussion for data in Supplementary Figure 3 could be located at Pg 6, line 18, 20, and 29.

- p10, line 35-p11, line 2. “Effective silencing of TRAIP markedly rescued cellular apoptosis, as demonstrated by the reduced PARP and caspase 3 cleavage, in H2170 cells by sustaining p-IkB without affecting the p53 acetylation (Figure S11C).” TRAIP expression doesn’t seem to be reduced in DMSO treated cells after siRNA treatment, is there a reason for this? The authors replied - “We agree with the Reviewer on this, and we are fully aware of this unusual observation. The experiment had been repeated (n = 3), and we consistently observed the expression of TRAIP protein in the DMSO-treated cells upon siRNA transfection. We reason that the presence of a compensatory mechanism could potentially stabilise TRAIP and counterbalance the knockdown effect, and this

mechanism may be impaired under cellular stress (eg drug treatment). However, this has not been explored further and the precise mechanism remains unclear. --- This statement should be introduced in the text.

We thank the Reviewer for the suggestion. The statement has been introduced in the text (Pg 11, line 36 – Pg 12, line 5) “It is interesting to note that TRAIP expression was consistently detected in the DMSO-treated cells upon siRNA interference, and we reasoned that the presence of a compensatory mechanism could potentially stabilize TRAIP and counterbalance the knockdown effect. We postulate that this mechanism could be impaired under cellular stress, thus allowing knockdown of TRAIP to be possible upon drug treatment. However, the detailed mechanism remains elusive at this moment.”

p12 lines 12-18. “The complex crosstalk between p53 and NFκB have been described previously, with the suppression of p65 in wild-type cells a result of competitive interaction with p300 or glucocorticoid receptor; conversely mutp53 prolongs and enhances NFκB activation. We propose here a novel mechanism for this divergent effect through acetylation of the DBD of mutp53R158G, which alters its DNA binding spectrum and upregulates TRAIP as key target gene, leading to NFκB suppression through TRAF2 degradation and culminating in cell death (Figure 7).” They need to do gel retardation to show DNA binding. We thank the Reviewer for the suggestion. Based on our understanding, cellular reporter assay and gel retardation assay (EMSA) are common techniques used to study protein-nucleic acid interaction, but EMSA assay is prone to false positive results. To address the DNA binding ability of mutp53, we have conducted reporter assays on mutp53R158G cells to indicate the binding of mutp53 to the promoter of CDKN1A (Supplementary Figure 4A) --- This is a reporter assay, a functional assay and not a DNA binding assay. They must do the DNA binding (gel retardation, for example and not the reporter assay) assay if they want to show DNA binding.

We agree with the reviewer that a gel retardation assay is essential to validate the DNA binding of mutp53 in our proposed model. Accordingly, we have performed electrophoretic mobility shift assay (EMSA) on both wtp53 and mutp53^{R158G}, expressed using *in vitro* translation system from the respective plasmid. In order to demonstrate that acetylation is crucial for the binding affinity of mutp53 to DNA, we also included the acetylated p53 isoforms (derived with the HAT domain of CBP/p300 with acetyl-CoA) in our EMSA. Our data showed that both wtp53 and mutp53^{R158G} could bind to the DNA oligo, which is a 30-bp p53 response element sequence from *CDKN1A*. Consistent our ChIP analyses, we detected stronger binding with EMSA when both wtp53 and mutp53^{R158G} are acetylated. These data are concordant with our claim that codon 158 mutp53 exhibit strong DNA binding affinity. The additional data have been included in Figure S4B-D, and described at Pg 6, line 35 and Pg 9, line 14.

Reviewer #2 (Remarks to the Author):

The revised manuscript is significantly improved although there are still a few points below that need to be addressed.

1. In rebuttal letter, the authors claimed that “It is likely that TRAIP is activated in these cell lines as a result of cisplatin-induced DNA intercalation through in a p53-independent manner, particular in null cells whereby the DNA-damage-repair axis is impaired.”

Unless the authors can see a much stronger induction of TRAIP by PXD+CDDP in p53R158G cells compared to p53 null cells (Fig. 5H), it is hard to tell if the induction of TRAIP is because of acetylated p53R158G or because of p53-independent factors. The authors should use multiple clones but not just a single clone to prove their hypothesis in Fig. 5H.

We agree with the reviewer on this. In order to increase the confidence on the specificity of TRAIPI-induced suppression of p-I κ B in p53^{R158G} cells, we have conducted the similar experiments on 5 independent p53^{R158G} clones derived from p53 null Calu-1 cells. Consistently, all 5 clones demonstrated a drop in I κ B phosphorylation that is associated with TRAIPI upregulation (Supplementary Figure 10). Densitometric quantification of TRAIPI expression has also been included in Figure S10B, suggesting that TRAIPI induction in p53^{R158G} cells is indeed stronger than that in null cells. These findings have been discussed in Pg 10, line 13-17.

2. Combination use of acetylation promoting reagents and DNA damage drugs in treating p53R158G tumor is the key point and major novelty of this manuscript, but the authors did not carry out any *in vivo* combination experiments to prove if this strategy works *in vivo*. The authors claimed that “belinostat exerts poor pharmacokinetic in mouse models with poor pharmacological effect on tumours (lack of histone acetylation).” Further analyses of the combination (single or combined) of these compounds and other DNA damage reagents on p53-mutant tumors may be useful.

We thank the reviewer for the emphasis on the combinatorial study, and we have conducted further experiments on two patient-derived xenograft models (GC-PDX and HCC-PDX). Topotecan is chosen to be used in combination with cisplatin based on the tolerability as well as the potency as a mutp53 acetylating agent. The *in vivo* data suggest that such combination may be beneficial in targeting codon 158 cancers, with significant tumor suppressive effect within a short window of treatment. In addition, topotecan and platinum-based chemotherapy have been proven to be well tolerated in a Phase III trial (J. Sehouli et al, 2016, Annals of Oncology), thereby strengthening its case as a potential therapeutic regime for *TP53* codon 158 cancers. These data have been presented in Figure 7J-K, and detailed in Pg 13, line 27-33.

REVIEWERS' COMMENTS:

Reviewer #2 (Remarks to the Author):

All the issues have been well addressed. This manuscript is acceptable for publication.